# Molecular profiling of brain endothelial cell to astrocyte endfoot communication in mouse and human

Steven A. Hill [1,2,3,4,13], Isabel Bravo-Ferrer [1,3,8,13], Austėja Čiulkinytė [1,2,3,4], Noelia Pérez Ramos [1,2,3,4], Ilaria Rossetti [1,2,3,4], Chiara Colvin[1,2,3,4], Paula Beltran-Lobo[1,2,3,4], Carlos Parra-Pérez [1,3,9,10,11], Katie Emelianova[1,3,12], Owen Dando [1,2,3,5], Beth Geary[6], Raja S. Nirujogi [6], Dario R. Alessi [6], Do-Young Lee[7], Youn-Bok Lee[7] & Blanca Díaz Castro [1,2,3,4] ✉

Our understanding of how the body communicates with the brain to coordinate their functions is remarkably limited. At the blood-brain barrier (BBB), brain endothelial cells (BECs) are ideally positioned to mediate signaling between blood and brain parenchyma via direct communication with astrocyte perivascular processes (endfeet). We develop a method to define the mouse in vivo astrocyte endfoot proteome, which in combination with BEC-specific RNA-seq, reveal BEC to astrocyte endfoot ligand-receptor pairs that are modulated when mice are exposed to a peripheral inflammatory insult with lipopolysaccharide. We show that over 80% of these mouse BEC-endfoot ligand-receptor pairs are also found in the human BBB, with a subset of them differentially expressed in human multiple sclerosis or Alzheimer's disease compared to healthy individuals. Our findings reveal dynamic BEC-endfoot communication pathways that are relevant to human physiology and provide methodology and datasets for the translational study of BEC-astrocyte cross-talk in health and disease.

Brain blood vessels maintain the brain microenvironment by tightly regulating the exchange of substances between the blood and the brain through a multicellular functional entity called the blood-brain barrier (BBB)[1]. Despite their essential role, our knowledge of how brain vascular cells communicate between themselves and with the cells in the brain parenchyma is remarkably limited. In capillaries, which form 85% of the brain vasculature[1,2], the BBB is composed of the brain endothelial cells (BECs), pericytes, astrocytes, and the basement

[1]Centre for Discovery Brain Sciences, The University of Edinburgh, Chancellor's Building, Edinburgh, United Kingdom. [2]Institute for Neuroscience and Cardiovascular Research, The University of Edinburgh, Chancellor's Building, Edinburgh, United Kingdom. [3]UK Dementia Research Institute at The University of Edinburgh, Chancellor's Building, Edinburgh, United Kingdom. [4]British Heart Foundation and UK Dementia Research Institute Centre for Vascular Dementia Research, Chancellor's Building, Edinburgh, United Kingdom. [5]Simons Initiative for the Developing Brain, University of Edinburgh, Edinburgh, United Kingdom. [6]Medical Research Council Protein Phosphorylation and Ubiquitylation Unit, School of Life Sciences, University of Dundee, Dundee, United Kingdom. [7]Department of Basic and Clinical Neuroscience, Maurice Wohl Clinical Neuroscience, Institute of Psychiatry, Psychology and Neuroscience, King's College London, London, United Kingdom. [8]Present address: Department of Cell and Developmental Biology, Division of Biosciences, University College London, London, United Kingdom. [9]Present address: Neurovascular Pathophysiology Group, Cardiovascular Risk Factor and Brain Function Programme, Centro Nacional de Investigaciones Cardiovasculares (CNIC), Madrid, Spain. [10]Present address: Unidad de Investigación Neurovascular, Departamento de Farmacología, Facultad de Medicina, Universidad Complutense de Madrid (UCM), Madrid, Spain. [11]Present address: Instituto Universitario de Investigación en Neuroquímica, Universidad Complutense de Madrid (UCM), Madrid, Spain. [12]Present address: Department of Botany and Biodiversity Research, University of Vienna, Vienna, Austria. [13]These authors contributed equally: Steven A. Hill, Isabel Bravo-Ferrer. ✉e-mail: b.diaz-castro@ed.ac.uk

membrane, which is the vascular extracellular matrix[3] (Fig. 1a). Astrocytes enwrap ~90% of the brain's vasculature surface through subcellular structures called astrocyte endfeet[4,5] (referred to as endfoot/endfeet hereon) while their cell body and thousands of leaflets in the brain parenchyma interact with neural cells[6–9]. Endfeet perform essential functions for the brain: they take up nutrients and hormones transported in the blood for processing and distribution to other

parenchymal cells[10,11], regulate local blood flow[12–15], enable the clearance of brain by-products[16], and contribute to the maintenance of a healthy BBB[17–19]. In capillaries, the abluminal side of the BECs faces both endfeet and pericytes, with most of its perimeter (~63%) adjacent to endfeet, separated from them only by the basement membrane[4,20–22]. Indeed, signaling from endfeet to BECs has been shown to be essential for the execution of several endfoot functions[23–26]. In contrast, and

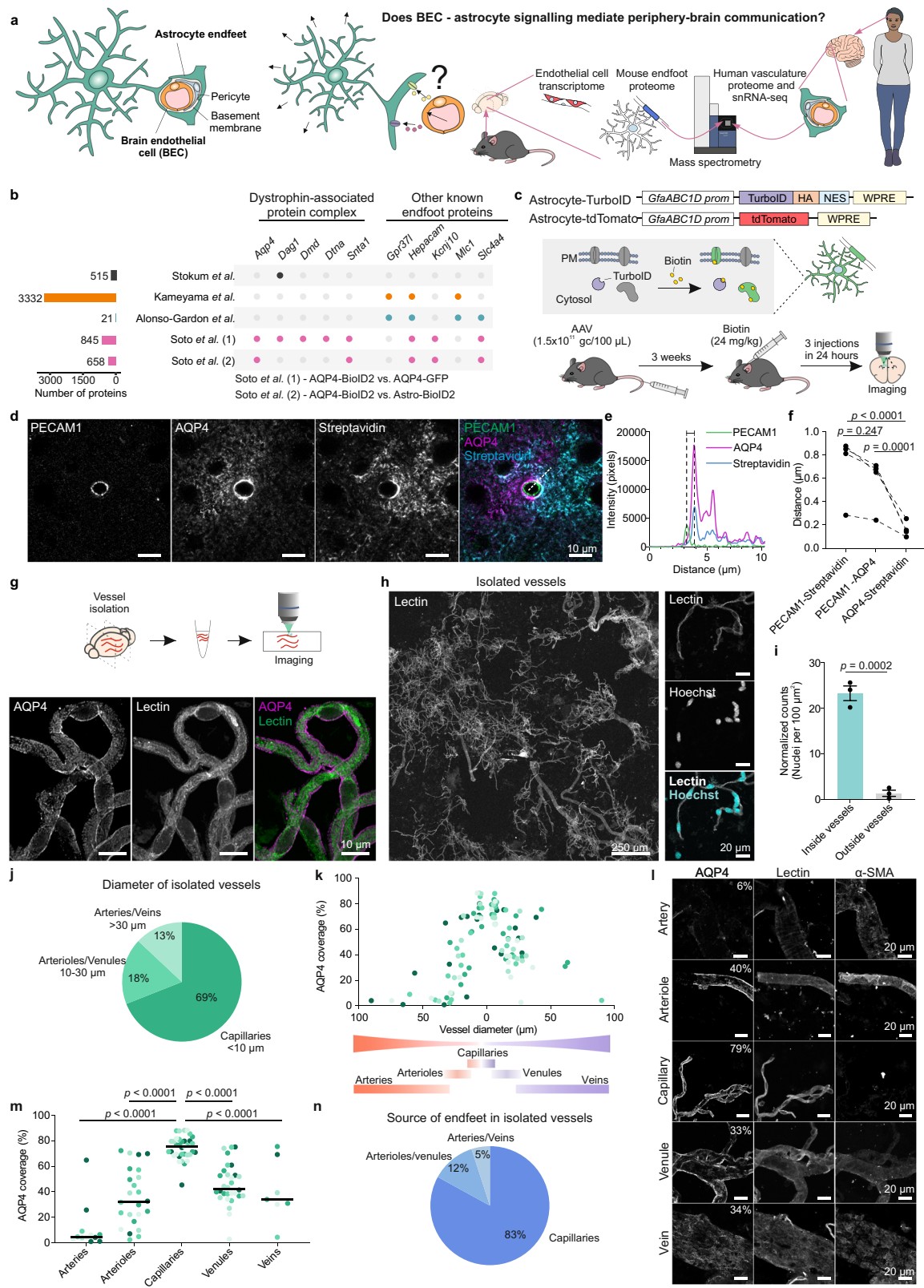

**Nature Communications** | (2025)16:9750

**Fig. 1 | Establishment of methods to identify astrocyte endfoot proteins.**
**a** Schematic of study aims. **b** Dot plot illustrating the endfoot-enriched dystrophin-associated protein complex (left) and other known endfoot proteins (gene names, right) in published endfoot proteome attempts[36–39]. Colored dots indicate presence of the protein, gray dots indicate absence (Stokum et al,. whole brain, black; Kameyama et al., whole brain, orange; Alonso-Gardon et al., whole brain, teal; Soto et al., striatum, pink). Horizontal bars indicate total proteins per dataset. **c** Top: AAV constructs generated and TurboID mechanism. GfaABC$_1$D prom: minimal GFAP promoter, HA: Human Influenza Hemagglutinin tag peptide, NES: nuclear export signal, WPRE: Woodchuck Hepatitis Virus Posttranscriptional Regulatory Element. Bottom: Schematic of AAV expression protocol. **d–f** Line analysis demonstrating biotinylated proteins (streptavidin) in endfeet (AQP4) vs brain endothelial cells (BECs; PECAM1) in Astrocyte-TurboID injected mice. Example 10 μm dotted line (**d**) and its intensity profiles (**e**) shown. **f** Quantified distance between PECAM1-streptavidin, PECAM1-AQP4, and AQP4-streptavidin. A linear mixed-effects model (LMM) was used to account for repeated measures per mouse: Measurement~Distance + (1|Animal), N = 4 mice, 3 images/mouse; F(2,30) = 23.638, Tukey post hoc. Dots are average per mouse. Dotted lines connect measurements by mouse. Scale bar = 10 μm. **g** Vessel isolation schematic (top) and representative magnified image of isolated blood vessels with endfeet (AQP4, magenta) around the vasculature (lectin, green). Scale bar = 10 μm. **h** Representative tile-scan image of isolated microvessels stained with lectin (vessels) and Hoechst for nuclei (cyan). Scale bars = 250 μm (left) and 20 μm (right). **i** Normalized Hoechst+ nuclei count. N = 3 images from one vessel isolation: Unpaired two-tailed t-test, (t(4) = 12.59). Data are mean ± SEM. **j** Proportional composition of vessel sizes and types (grouped by diameter) in isolated microvessels calculated as area fraction of the total vessel area in (**h**). **k** Percent AQP4 coverage (fraction of lectin+ vessel area colocalizing with AQP4) by vessel diameter. Dots left of 0 indicate arteries/arterioles; right of 0, venules/veins. Data points are shaded by mouse. N = 108 vessels from 5 mice, 19–23 vessels/mouse. Diagrams below represent continuum of vessel diameter/type. **l** Representative images of vessels in (**k**) stained with AQP4, lectin, and alpha-smooth muscle actin (α-SMA; striated smooth muscle lining arterioles/arteries). Percent AQP4 coverage indicated in the first column. Scale bars = 20 μm. **m** Percent AQP4 coverage by vessel type. Horizontal line indicates median, and data points are shaded by mouse. LMM was used to account for repeated measures per mouse: Measurement~Vessel_type + (1|Animal), N = 108 vessels from 5 mice, 19–22 vessels per mouse with 1–8 vessels/category/mouse; F(4,100.69) = 41.664, Tukey post hoc. **n** Proportion of endfeet isolated from each vessel type. Exact sample sizes per animal and source data are provided in a Source Data file.

likely critical for periphery-brain communication, our knowledge of BEC to astrocyte signaling remains remarkably unexplored despite some in vitro[27–29] evidence and recent emergence of data supporting it could also take place in vivo[30–32]. BEC to astrocyte signaling is of great interest particularly in the context of peripheral infections, which are known risk factors for stroke, delirium and dementia[33–35], with unclear underlying mechanisms that lead to cognitive decline.

Here, we hypothesized that BECs continuously signal to astrocytes to maintain proper brain homeostasis and that these interactions are dynamic depending on events that occur in the periphery (Fig. 1a). To address this in vivo, we developed a method that combines protein TurboID-mediated biotinylation with vessel purification to identify the mouse endfoot proteome, which, together with the BEC-specific translatome, allowed for the identification of over 300 ligand-receptor pairs potentially involved in the signaling of BECs to astrocytes. We discovered that these interactions were modulated by peripheral events, i.e., signaling pathways uniquely appear under peripherally induced inflammation. In addition, a large proportion of the mouse ligand-receptor pairs are present in the human BBB with a fraction of them being altered in human multiple sclerosis (MS) and Alzheimer's disease (AD). Overall, our data suggests that the BEC-endfoot signaling axis could be an important mediator of periphery-brain communication that is relevant to human health and disease.

## Results

### Astrocyte-TurboID to label and purify astrocyte and endfoot proteins

To identify endfoot receptors potentially involved in communication with BECs, we first sought to robustly define the proteome of this subcellular compartment. While previous efforts have provided useful insights[36–39], they were limited, either by defining the interactome of a single protein enriched in but not restricted to the endfoot (hepacam[36] or AQP4[38]), or by the lack of definitive evidence supporting a complete and pure endfoot isolation after biochemical purification[37,39]. As a result they only identify a subset of well-known endfoot enriched proteins[5,40], i.e. AQP4 (*Aqp4*[41]) and the dystrophin-associated protein complex (DAPC) that helps anchor AQP4 to the plasma membrane (dystrophin (*Dmd*[42]), α-syntrophin (*Snta1*[43]), α-dystrobrevin (*Dtna*[44]), and β-dystroglycan (*Dag1*[45])), in addition to *Gpr37l1*[36], *Hepacam*[46], *Kcnj10*[47], *Mlc1*[48], *Slc4a4*[49] (Fig. 1b). To overcome the limitations of previous studies, we combined astrocyte specific proximity biotinylation (mediated by TurboID[50]) with vessel purification and liquid chromatography tandem mass spectrometry (LC-MS/MS). Using adeno-associated viruses (AAVs) to express brain-wide[51] cytosolic TurboID (Astrocyte-TurboID) or the control tdTomato (Astrocyte-tdTomato) under an astrocyte specific promoter (GfaABC$_1$D; Fig. 1c), we achieved astrocyte-specific protein biotinylation in the cortex of mice with Astrocyte-TurboID compared to Astrocyte-tdTomato (Supplementary Fig. 1). Around the vasculature, biotinylation was also highly specific to endfeet and not found in BECs (Fig. 1d–f).

Building on previous findings demonstrating that endfeet remain attached to blood vessels isolated from the brain parenchyma[52,53], we reasoned that isolating brain blood vessels from mice injected with Astrocyte-TurboID AAV would allow us to subsequently purify the biotinylated proteins found in the endfeet. We therefore isolated cortical vessels and confirmed that they retain endfeet (Fig. 1g) and present negligible non-vascular cell contamination (Fig. 1h, i). The vessel preparations contained vessels of different sizes with a strong enrichment of capillaries (69% of the total vessel area in the image; Fig. 1h, j). Among the different vessel types, the coverage of endfeet measured by AQP4 signal was distinct, with the highest endfoot coverage found on capillaries (Fig. 1k–m). By multiplying the percentage of vessels of certain diameter (Fig. 1j) with the mean of their endfoot coverage (Fig. 1m), we estimated that 83% of the endfoot content in purified vessels belong to capillaries (Fig. 1n), the vascular segments in which endfeet and BECs could directly communicate with each other. Thus, we purified vessels from cortices of Astrocyte-TurboID or Astrocyte-tdTomato injected mice, homogenized them for streptavidin-mediated pulldown (Fig. 2a, bottom), and identified the purified proteins through LC-MS/MS (Fig. 2a, b). We also purified and identified cortical astrocyte proteins, by skipping the vessel isolation step (Fig. 2a, top). We determined the astrocyte and endfoot proteomes by comparing proteins from mice injected with Astrocyte-TurboID versus the negative control Astrocyte-tdTomato AAVs (Fig. 2c–g). The astrocyte proteome consisted of 393 TurboID-unique (Fig. 2d) plus 1658 TurboID-enriched proteins versus the tdTomato samples (Fig. 2e and Supplementary Data 1). Similarly, we defined the endfoot proteome consisting of 2293 proteins (Fig. 2f, g and Supplementary Data 2). We confirmed that the proteins identified were astrocyte-specific by assessing the enrichment of known cell markers in our datasets (Supplementary Fig. 2a). To further assess the specificity of the endfoot proteome, we focused on dystrophin, a protein with multiple cell- and organ-specific isoforms that are encoded by a large gene with multiple promoters[54]. It is well described that the dystrophin Dp71 isoform is enriched in endfeet[55,56]. Using peptide data from our endfoot proteomics analysis, we mapped the detected peptides onto the full dystrophin gene (*Dmd*) sequence. This revealed a clear enrichment in the region corresponding to the Dp71 isoform, further supporting the selective enrichment of endfoot proteins in our preparation (Supplementary Fig. 2b, c). A few peptides aligned with the

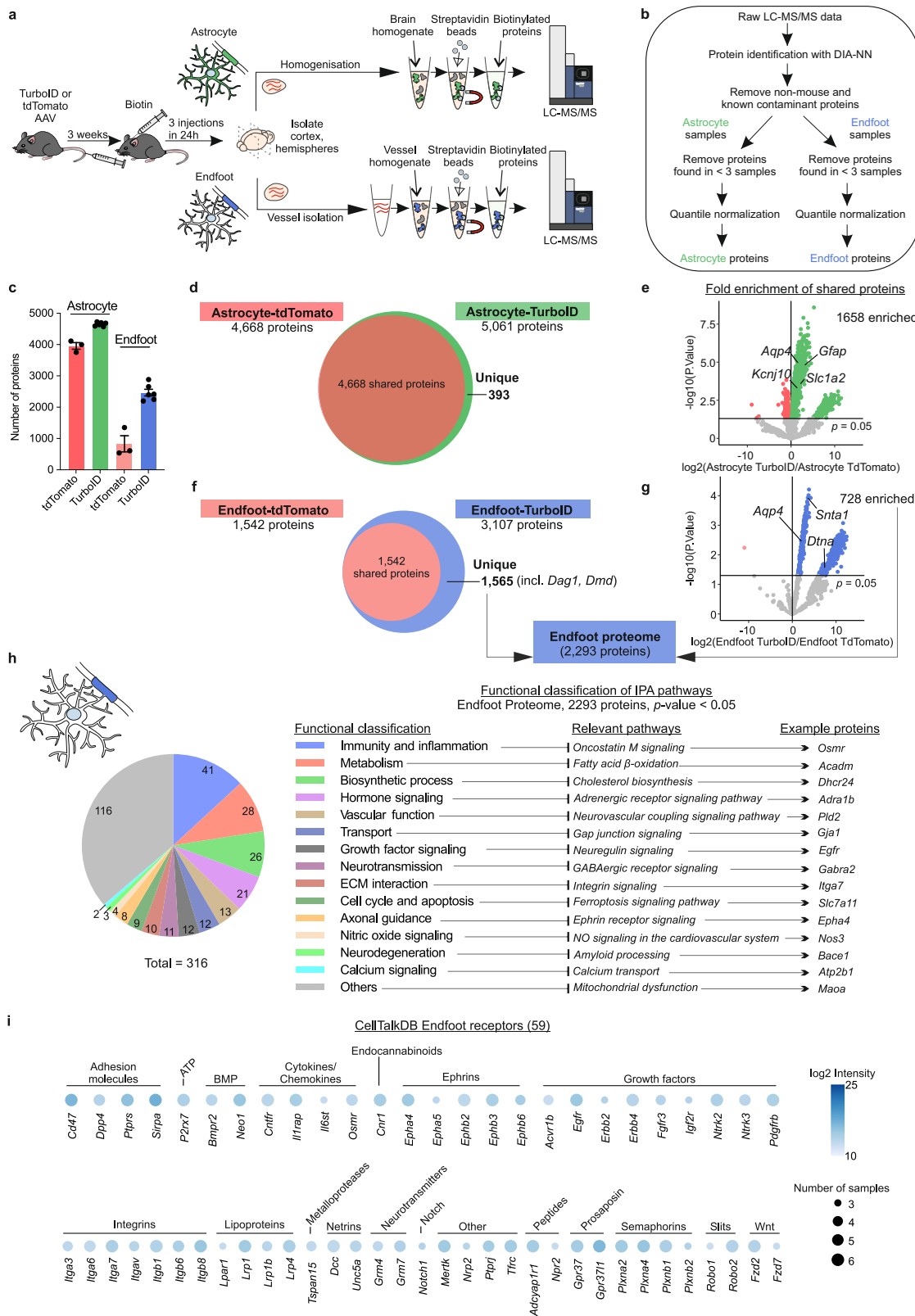

**h** Functional classification of IPA pathways
Endfoot Proteome, 2293 proteins, p-value < 0.05

| Functional classification | Relevant pathways | Example proteins |
|---|---|---|
| Immunity and inflammation | Oncostatin M signaling | Osmr |
| Metabolism | Fatty acid β-oxidation | Acadm |
| Biosynthetic process | Cholesterol biosynthesis | Dhcr24 |
| Hormone signaling | Adrenergic receptor signaling pathway | Adra1b |
| Vascular function | Neurovascular coupling signaling pathway | Pld2 |
| Transport | Gap junction signaling | Gja1 |
| Growth factor signaling | Neuregulin signaling | Egfr |
| Neurotransmission | GABAergic receptor signaling | Gabra2 |
| ECM interaction | Integrin signaling | Itga7 |
| Cell cycle and apoptosis | Ferroptosis signaling pathway | Slc7a11 |
| Axonal guidance | Ephrin receptor signaling | Epha4 |
| Nitric oxide signaling | NO signaling in the cardiovascular system | Nos3 |
| Neurodegeneration | Amyloid processing | Bace1 |
| Calcium signaling | Calcium transport | Atp2b1 |
| Others | Mitochondrial dysfunction | Maoa |

Total = 316

Dp427 isoform, which has been shown to be expressed in arterioles[57]. This is concordant with the presence of a fraction of arteriolar endfeet in our protein purification (Fig. 1k–n).

Further comparison of our endfoot proteome to previous datasets demonstrated that, while a large proportion of other endfoot datasets was represented in our data (Supplementary Fig. 3a and Supplementary Data 3), ours is the only dataset in which all

components of the DAPC, along with other well-known endfoot proteins, are enriched when compared to both the tdTomato control (proteins present in endfeet) and the whole astrocyte (proteins enriched in the endfeet vs the rest of the astrocyte; Supplementary Fig. 3b and Supplementary Data 2 and 3). This, together with the data in Figs. 1 and 2, and Supplementary Fig. 2, indicates that our method provides the most precise endfoot proteome so far.

**Fig. 2 | Proteomics profiling of the astrocyte endfoot using Astrocyte-TurboID.** **a** Schematic of protocol for proteomics experiments. Mice received either Astrocyte-TurboID or Astrocyte-tdTomato AAVs as well as biotin injections. Cortices were isolated and, from the same mouse, one hemicortex was used to identify the astrocyte proteome and the other for the endfoot proteome. **b** Flowchart of LC-MS/MS data analysis for protein identification. **c** Bar graph of the number of proteins found in each sample type. Each dot represents one mouse hemibrain ($N = 6$ for TurboID, $N = 3$ for tdTomato; data are represented as mean ± SEM). **d** Venn diagram comparing proteins found in Astrocyte-TurboID and Astrocyte-tdTomato samples. **e** Volcano plot demonstrating the fold enrichment of shared proteins between Astrocyte-TurboID and Astrocyte-tdTomato. Several known astrocyte proteins are labeled. 1658 proteins were enriched in Astrocyte-TurboID. **f** Venn diagram comparing proteins found in Endfoot-TurboID and Endfoot-tdTomato samples. **g** Volcano plot demonstrating the differential expression of shared proteins between Endfoot-TurboID and Endfoot-tdTomato. Several known endfoot proteins are labeled. 728 proteins were enriched in Endfoot TurboID. The 728 enriched + 1565 unique proteins in Endfoot TurboID comprise the endfoot proteome (2293 proteins). **h** Functional classification of pathways represented by endfoot proteins. Pathways were identified using Ingenuity Pathway Analysis and curated into the categories listed. **i** Dot plot for receptors listed in CellTalkDB and found in the endfoot proteome. The type of ligand for each receptor is indicated above the dots. "Other" refers to receptors with multiple ligand types. The color intensity of the dot corresponds to the abundance of the protein, while the size of the dot corresponds to the number of samples in which the protein was identified (total $N = 6$ mice). Proteins are referred to by their gene names. Differential expression was performed using Limma. Source data are provided as a Source Data file and in Supplementary Data 1 and 2.

## The endfoot proteome reveals proteins involved in molecular transport, metabolism and cell-cell communication

To expand our understanding of the biological functions of endfeet, we used Ingenuity Pathway Analysis (IPA)[58] and identified a total of 316 pathways significantly represented by the endfoot proteome ($p$-value < 0.05) (Fig. 2h and Supplementary Data 2). Curated categorization of these pathways highlighted the role of endfeet in uptake of molecules from the blood into the brain ("Transport") and their potential to metabolize these molecules ("Metabolism", "Biosynthetic process"). Accordingly, we identified 128 solute transporters in our endfoot proteome (Supplementary Fig. 4), many of them involved in the uptake of nutrients like glucose (*Slc2a1*/GLUT-1), monocarboxylates lactate/pyruvate (*Slc16a1*/MCT-1, *Slc16a3*/MCT-4, *Slc16a7*/MCT-2), acetyl-CoA (*Slc33a1*), amino acids (*Slc1a1*/EAAT3, *Slc1a2*/EAAT2, and *Slc1a3*/EAAT1), and, in the mitochondria, nucleotides (*Slc25a3*, *Slc25a4*, *Slc25a5*), vitamins (*Slc25a19*, *Slc25a32*), or fatty acids (*Slc25a20*). Among the most characteristic astrocyte metabolic functions is the storage and mobilization of glycogen to support neuronal function[11,59], with glycogen accumulation observed in the endfeet[20,60]. Accordingly, our endfoot proteome contains the enzymes required for glycogenesis (glycogen synthase, *Gys1*, and glycogen synthase kinase 3 alpha and beta, *Gsk3a* and *Gsk3b*), and glycogenolysis (glycogen phosphorylases, *Pygb* and *Pygm*) (Supplementary Data 2). To corroborate this finding, we performed immunofluorescence quantification and confirmed expression of PYGB in endfeet, along with its depletion in vascular cells, in both mouse and human cortex (Supplementary Fig. 5) – indicating that the proteins we identify in this dataset may be conserved across species.

Many of the pathways identified in the endfoot proteome relate to immunity and inflammation, hormone signaling, or vascular function, suggesting endfeet contain proteins able to detect and respond to external cues (Fig. 2h). We further identified 59 receptors with known ligands and known downstream signaling cascades using the CellTalkDB database[61], which underline the potential of endfeet to receive signals such as chemokines and cytokines, growth factors, lipoproteins, and other secreted proteins like WNT (Fig. 2i). Overall, these results reveal that endfeet contain a multitude of proteins capable of receiving molecules at the BBB.

## The expression of BEC secreted proteins is altered after peripherally induced inflammation

The identification of inflammation-related signaling in endfeet (Fig. 2h, i) and in vitro data showing that conditioned media from cultured BECs exposed to the bacterial endotoxin lipopolysaccharide (LPS) induces reactive phenotypes in astrocytes[29] led us to hypothesize that BECs sense circulatory signals and release ligands to relay the peripheral inflammation to endfeet prompting astrocyte immune responses. To test this, we used an established model of peripherally induced inflammation in which mice receive a single intraperitoneal injection of LPS[62]. First, we defined the cortical BEC responses in vivo by

administering LPS to *Cdh5-Cre/ERT2*::Ribotag mice[63,64], or PBS as a control, and performed BEC-specific RNA sequencing (RNA-seq; Fig. 3a). After confirming with immunofluorescence and gene expression analyses that the sequenced transcripts were specific to BECs (Supplementary Fig. 6a–c), we compared BEC gene expression between LPS- and PBS-injected mice and identified over 3500 differentially expressed genes (DEGs) (Supplementary Fig. 6d and Supplementary Data 4). We found that BECs acquire a prominent immune role (Fig. 3b–d and Supplementary Data 4) together with changes in hormone signaling, vascular function, and transport. Among the DEGs, we identified many that encode secreted proteins. We identified 83 ligands that were upregulated after LPS, including cytokines, chemokines, and other immune signaling mediators (Fig. 3d). We also found 87 downregulated ligands, which included growth factors and extracellular matrix components, suggesting a downregulation of homeostatic genes in response to acute inflammation (Supplementary Fig. 6e). Overall, our data suggest BEC communication with nearby cells is altered by LPS.

## Communication pathways between BECs and endfeet are modulated by LPS

We next asked whether the endfoot proteome was affected by peripherally induced inflammation, after confirming that LPS injection had no effect on TurboID-mediated protein biotinylation in astrocytes or the endfoot coverage of purified vessels (Supplementary Fig. 7a–j). We identified 56 unique or differentially expressed endfoot proteins between LPS and PBS mice (Fig. 3e, f and Supplementary Data 5). Classification of the differentially expressed proteins into their main functions and pathways (Fig. 3g, h) indicated that LPS activates endfoot proteome remodeling by modulating protein degradation, translation or transcription. Other molecular modifications included cytoskeleton remodeling, molecular transport, and cell adhesion. In addition, several cell-cell communication pathways were predicted to be activated with LPS, such as HER-2 (ERBB2), interleukin-1, semaphorin, EPH-Ephrin or ROBO signaling (Fig. 3h and Supplementary Data 5). These data demonstrate that at 24 h post-LPS, the endfoot undergoes proteome remodeling that influences intracellular functions and its interactions with the surrounding cells.

To identify BEC-endfoot communication pathways and assess if this communication is modulated by LPS, we combined the LPS and PBS endfoot proteomes (Supplementary Fig. 7k–l and Supplementary Data 5), and using the CellTalkDB database[61] identified which ligands found in BECs had a corresponding receptor expressed in endfoot (Fig. 4a). First, we looked for ligand-receptor pairs observed in PBS samples and identified 360 BEC-endfoot ligand-receptor pairs, indicating these cells are actively communicating in healthy conditions (Fig. 4b, gray circle). Importantly, analysis of LPS-injected mice revealed that the relationship between BEC and endfeet is dynamic, with 33 ligand-receptor pairs downregulated after LPS (Fig. 4b purple circle, and Fig. 4c) and 53 upregulated (Fig. 4b red circle, and Fig. 4d),

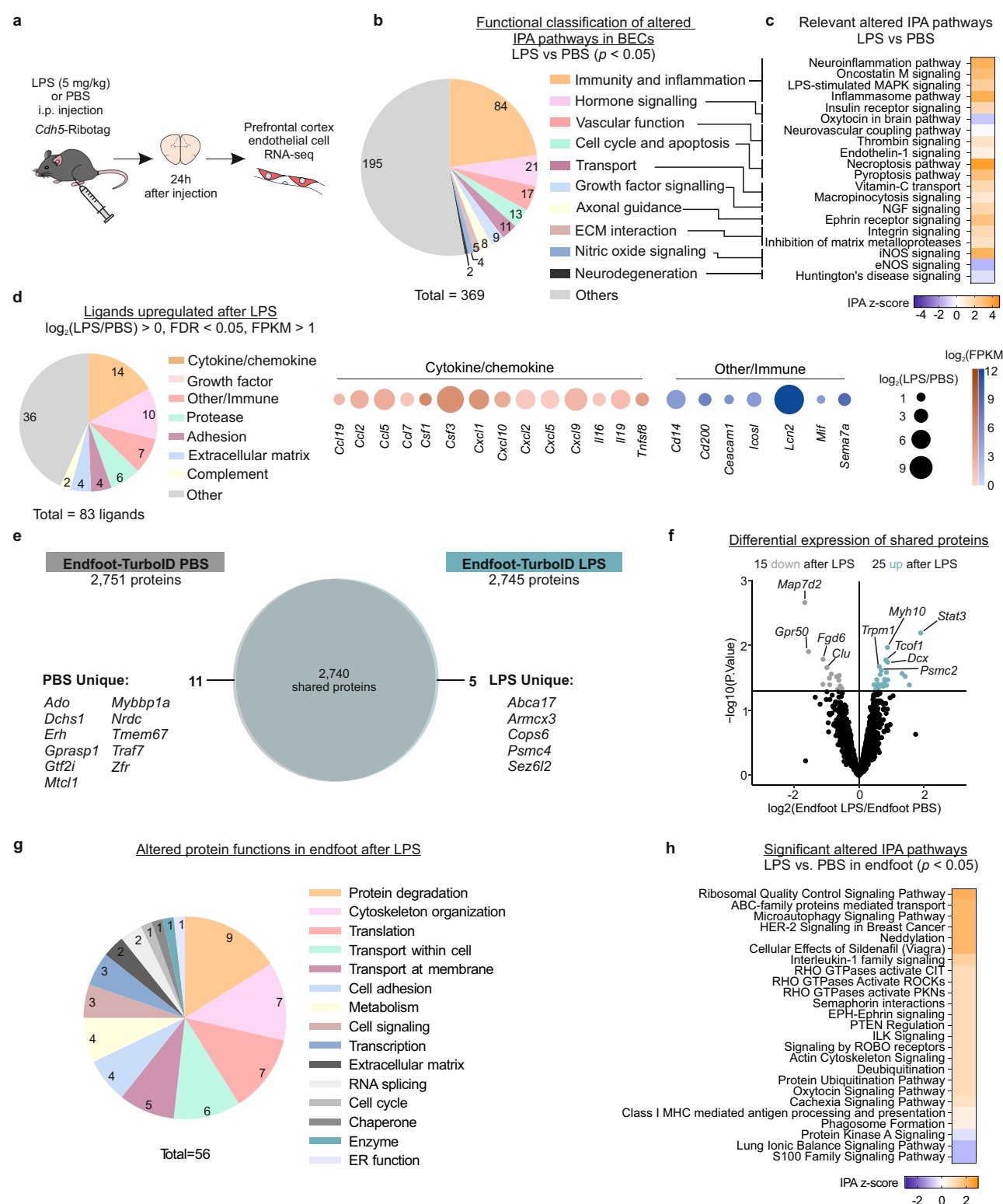

including 11 "induced" that were not found in PBS injected mice (Fig. 4b red circle, and Fig. 4e). Unexpectedly, just a few of the ligand-receptor pairs altered by LPS were conventional immune-response related pathways, such as SPP1-CD44, LCN2-SLC22A17, and CXCL10-DPP4. In addition to these, we found others relevant to adhesion (e.g., integrins) and classic cell-cell communication pathways mediated by plexins, NOTCHs, or WNTs (Fig. 4c–e). Furthermore, several of the identified ligand-receptor pairs mediate signaling pathways identified as altered in endfeet of LPS-injected mice compared to their PBS controls, like semaphorin, ROBO receptor, and HER-2 (ERBB2)

signaling (Figs. 3h and 4c, d), indicating that BEC-endfoot communication could have downstream effects in endfeet. With these results, we demonstrate that BECs have the capacity to signal to endfeet in a context-dependent manner to relay information from the periphery into the brain parenchyma.

### WNT10B-FZD7 as a pathway for BEC-endfoot communication

An intriguing ligand-receptor pair induced by LPS was WNT10B-frizzled7 (FZD7; Fig. 4e). WNT signaling has been shown to be particularly important for angiogenesis, vascular development, and BBB

**Fig. 3 | BEC and astrocyte endfoot molecular changes during peripherally induced neuroinflammation. a** Schematic of brain endothelial cell (BEC) RNA-seq workflow. **b** Functional classification of altered pathways after LPS administration from Ingenuity Pathway Analysis with curated categories. **c** Example IPA pathways within each category and their predicted activation (orange) or inhibition (blue) IPA z-scores. **d** (Left) Classification of CellTalkDB ligands upregulated after LPS (total = 83 ligands). (Right) Dot plots for cytokine/chemokine ligands and other immune-related ligands upregulated after LPS. For dot plots, the color intensity of the dot corresponds to the abundance of the gene in LPS samples (Log$_{10}$FPKM), while the size of the dot corresponds to the log$_2$(LPS/PBS) of each gene. Proteins are referred to by their gene names. **e** Venn diagram comparing proteins found in

the endfoot proteome of PBS-injected control mice (Endfoot-TurboID PBS, gray circle) and LPS-injected mice (Endfoot-TurboID LPS, teal circle). Unique proteins from each condition are indicated. **f** Volcano plot demonstrating the differential expression of shared proteins between Endfoot-TurboID LPS and Endfoot-TurboID PBS samples. 15 proteins were downregulated after LPS, and 25 proteins were upregulated. Differential expression was performed using Limma. **g** Functional classification of proteins altered by LPS (unique and enriched proteins, both up and down after LPS). Protein function was derived from Uniprot and curated into the categories listed. **h** IPA pathways significantly altered by LPS and their predicted activation (orange) or inhibition (blue) IPA z-scores. Source data are provided in Supplementary Data 4 and 5.

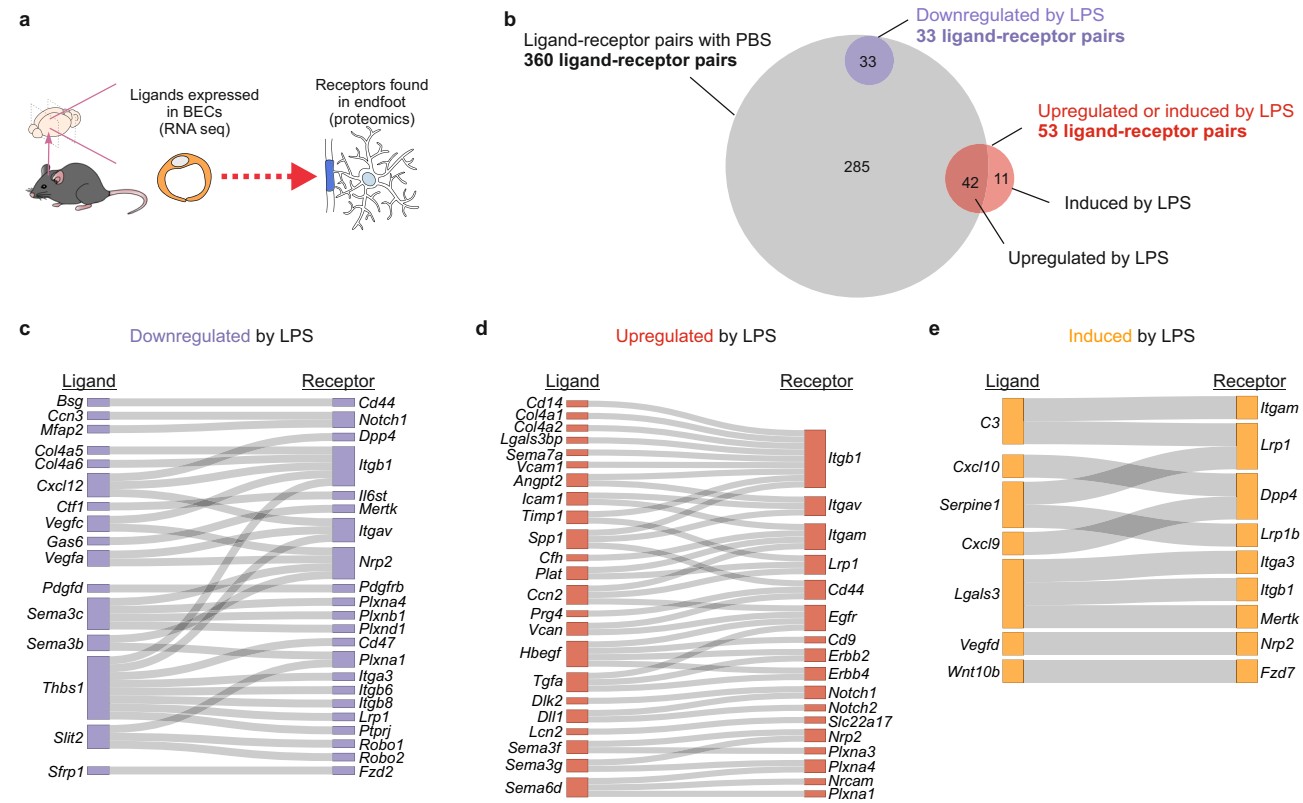

**Fig. 4 | Ligand-receptor pairs between BECs and astrocyte endfeet and their modulation during peripheral inflammation. a** Diagram of the search criteria for ligand-receptor pair identification. Ligand-receptor pairs were considered when ligands were found in BEC RNA-seq data and a corresponding receptor was found in the endfoot proteome. **b** Venn diagram showing overlap of identified ligand-receptor pairs from mice at baseline (gray, PBS BEC IP > 1 FPKM and PBS endfoot

proteome), downregulated by LPS (purple, BEC log$_2$(LPS/PBS) < −1, FDR < 0.05 and PBS + LPS endfoot proteome), or upregulated by LPS (red, log$_2$(LPS/PBS) > 1, FDR < 0.05 and PBS + LPS endfoot proteome). **c-e** Sankey plots illustrating interactions between ligand-receptor pairs found to be downregulated (**c**), upregulated (**d**), or induced by LPS (i.e., only present in the LPS condition; **e**) Source data are provided in Supplementary Data 7.

maintenance[23,65]. However, to our knowledge, it has never been described to occur from BECs to endfeet. In line with sepsis and peripheral infections increasing the risk for stroke[35], polymorphisms of *WNT10B*[66] are associated with Korean cerebral infarction; and *Fzd7* activation (in a yet-to-be-identified cell type) has been shown to attenuate BBB disruption after stroke via a downstream signaling cascade involving dishevelled and catenin beta-1[67]. Encouraged by this evidence, we further investigated the potential involvement of WNT10B-FZD7 as a means for BEC-endfoot communication to contribute to periphery-brain communication upon LPS administration. We validated the expression of FZD7 in endfeet with immunohistochemistry of expanded cortical brain tissue (using CUBIC-X[68]) from wild type mice that received GfaABC$_1$D-GFP AAVs to express green fluorescent protein (GFP) in astrocytes (Fig. 5a). Quantification of FZD7 signal overlap with GFP-positive endfeet and lectin-labeled vessels in capillary cross-sections confirmed FZD7 presence in endfeet and its

scarcity in the vasculature (Fig. 5b). To further validate these results, we isolated cortical vessels and stained them for FZD7, AQP4, and lectin (Fig. 5c). Signal colocalization analyses confirmed that FZD7 coincides with the endfoot marker AQP4 and not lectin in purified vessels as well (Fig. 5c, d, f, g). Consistently, enzymatic endfoot depletion of isolated vessels led to the loss of both AQP4 and FZD7 signal, while lectin remained intact, further demonstrating that FZD7 is localized in endfeet (Fig. 5c–e, h, i). Finally, in agreement with the proteomics data, protein quantification analyses of FZD7 within the endfoot area showed no differences in its expression after LPS when compared to PBS controls (Fig. 6a–d).

We next assessed the expression of WNT10B through RNA in situ hybridization (RNAscopeTM) (Supplementary Fig. 8a, b) and immunohistochemistry (Fig. 6e), confirming that *Wnt10b* RNA is found in cortical-capillary BECs and its resultant protein in capillaries. Both RNA and protein quantifications demonstrated an increase of

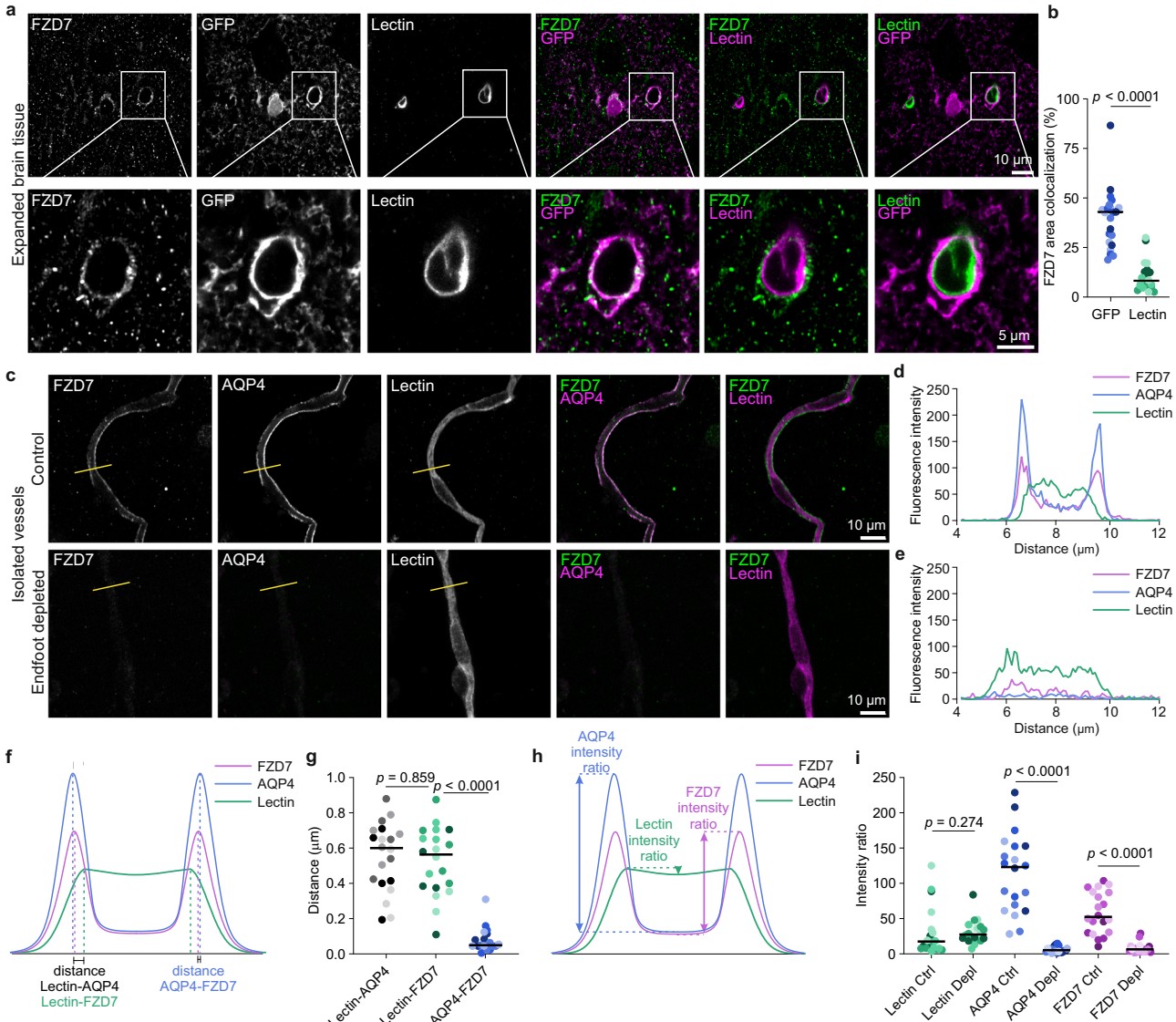

**Fig. 5 | Validation of FZD7 localization in astrocyte endfeet. a** Representative CUBIC-X expanded immunofluorescence images of FZD7, GFP-labeled astrocytes with endfeet, and lectin-labeled vessels in cortex of wild-type mice injected with AAV PHP-GfaABC$_1$D-GFP. Expansion ratio (ratio measurement of astrocyte territory) = 1.8×. White squares indicate region of each image displayed at higher magnification below. Scale bars = 10 μm (overview) and 5 μm (inset). **b** Percent area of endfoot-GFP or lectin signal colocalizing with FZD7. Data shown as individual values, data points are shaded by mouse, and horizontal lines represent the median. LMM was performed to account for repeated measures per mouse: % Area-Marker + (1|Animal), $N = 22$ vessels from 4 mice, 5 or 6 vessel cross-sections/ mouse; $F_{(1,38.81)} = 74.55$. **c** Representative images of isolated cortical capillaries with endfeet (top) or endfoot depleted (bottom) and labeled for FZD7, AQP4 and lectin. Scale bar = 10 μm. Yellow line indicates position for pixel intensity profiles in (**d**, **e**). **d**, **e** Representative intensity profiles of FZD7 (magenta), AQP4 (blue), and Lectin (green) for isolated vessels with endfeet (**d**) and endfoot-depleted vessels (**e**). **f** Schematic of peak-to-peak distance measurement between peak intensities of

FZD7 (magenta), AQP4 (blue) and Lectin (green). **g** Distance (μm) between peak intensities: Lectin-AQP4 (gray shades), Lectin-FZD7 (green shades), and AQP4-FZD7 (blue shades). Data are individual values, data points are shaded by mouse, and horizontal lines represent the median. LMM was performed to account for repeated measures per mouse: Distance-Marker + (1|Animal), $N = 4$ mice, 5 images/ mouse; $F_{(2,54)} = 60.08$, Tukey post hoc. **h** Schematic of intensity ratio metric (edge vs. center of vessel) for FZD7 (magenta), AQP4 (blue) and Lectin (green). **i** Intensity ratios of Lectin (green), AQP4 (blue), and FZD7 (magenta) in isolated vessels with endfeet and endfoot-depleted samples. Data are individual values, data points are shaded by mouse, and horizontal lines represent the median. LMM was performed on the indicated pairs to account for repeated measures per mouse: Intensity ratio-Condition + (1|Animal), $N = 20$ images/condition from 4 control and 5 depleted mice, 2-5 images/mouse. Lectin Ctrl vs. Depl, $F_{(1,7.38)} = 1.39$; AQP4 ctrl vs. Depl, $F_{(38)} = 191.84$; FZD7 ctrl vs. FZD7 Depl, $F_{(38)} = 54.85$. Exact sample sizes per animal and source data are provided in a Source Data file.

WNT10B in the vessels of LPS samples (Fig. 5e–h and Supplementary Fig. 8a–d), consistent with its induction in the RNA-seq experiment (Fig. 4e and Supplementary Data 4). In addition, we investigated the relevance of this ligand-receptor pair to human vasculature with immunofluorescence for FZD7 and WNT10B in frontal cortex of adult healthy subjects (39–45 years old). In humans, both FZD7 and WNT10B showed enrichment in capillaries (Supplementary Fig. 9). However, unlike mice, human brains displayed expression of FZD7

in vascular cells, in addition to the endfeet (Supplementary Fig. 9a, b).

We further explored our datasets to determine if WNT10B-FZD7 upstream and downstream pathways were detected in BECs and endfeet, respectively, and if they were activated by LPS administration. *Wnt10b* gene expression is driven by the NF-κB complex[69], and LPS binding to its receptor TLR4 leads to NF-κB complex translocation to the nucleus[70] to control gene expression. Thus, we investigated the

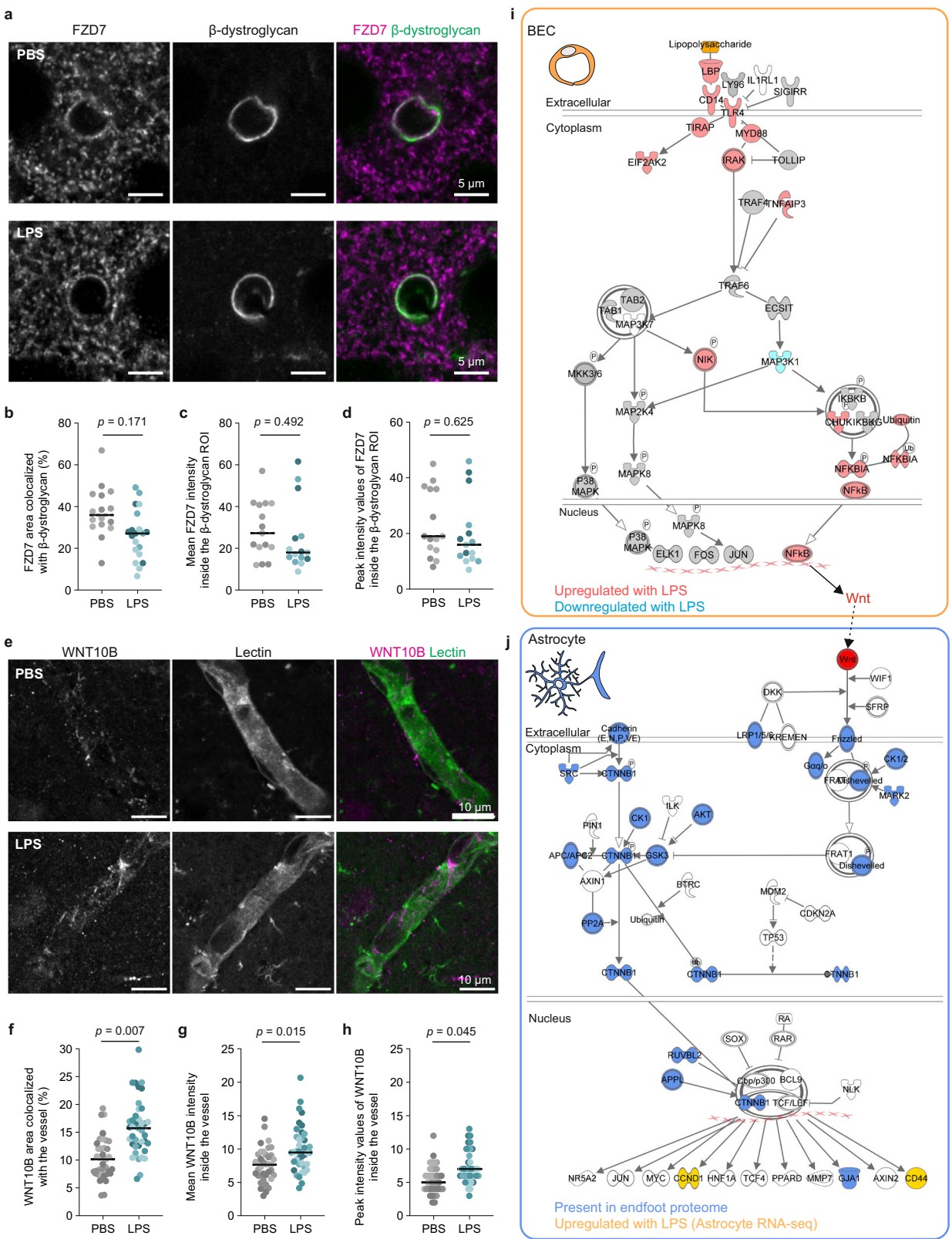

expression of the TLR4 pathway components in our BEC RNA-seq data and found that *Tlr4* and the majority of its interactors and downstream effectors, including all components of the NF-κB complex, were either upregulated (red shapes) or present (gray shapes) in BECs after LPS stimulation (Fig. 6i and Supplementary Fig. 10a), suggesting that LPS can directly activate BECs via the TLR4 pathway and induce NF-κB activity to prompt the expression of *Wnt10b* (Fig. 6e–i and

Supplementary Figs. 8 and 10a). Additionally, we found that endfeet have the machinery to respond locally to WNT ligands, with most of the proteins downstream of the FZD7 canonical pathway, including dishevelled and catenin beta-1, being present in the endfoot proteome (blue shapes; Fig. 6j and Supplementary Fig. 10b). Finally, using published astrocyte specific RNA-seq in the same LPS model[62], we confirmed that two gene targets of nuclear catenin beta-1 were

**Fig. 6 | WNT10B-FZD7 signaling between BECs and astrocyte endfeet during peripheral inflammation. a** Representative images of FZD7 (magenta) and endfeet (β-dystroglycan, green) in cortical capillaries of mice receiving PBS (top) or LPS (bottom). Scale bars = 5 μm. **b** β-dystroglycan+ area colocalizing with FZD7 24 h after PBS or LPS injection. $N$ = 16 images from 3 PBS and 20 images from 3 LPS mice, 5–8 images/mouse. FZD7 mean (**c**) or peak (**d**) intensity within β-dystroglycan region of interest (ROI) after LPS and PBS. $N$ = 15 images from 3 mice/treatment, 5 images/mouse. **e** Immunofluorescence of WNT10B (magenta) and lectin (green) in cortical capillaries of mice receiving PBS (top) or LPS (bottom). Scale bars = 10 μm. **f** Percent area of blood vessel (ROI that contained both endfeet and the vessel by using β-dystroglycan staining to define the endfoot boundary) colocalizing with WNT10B 24 hours after PBS or LPS injection. WNT10B mean (**g**) or peak (**h**) signal intensity after LPS and PBS 24 h after LPS or PBS injection. LMM was used to account for repeated measures from each mouse: Measurement-Treatment + (1| Animal). For (**b**), $F_{(1,3.91)}$ = 2.81; for (**c**), $F_{(1,4)}$ = 0.571; for (**d**), $F_{(1,4)}$ = 0.280. For (**f**–**h**), $N$ = 39 images/treatment from 4 LPS mice and 4 PBS mice, 10 vessels/mouse;

for (**f**), $F_{(1,5.9)}$ = 16.65; for (**g**), $F_{(1,5.96)}$ = 11.63; for (**h**), $F_{(1,6.03)}$ = 6.40. Horizontal lines represent the median; data points are shaded by mouse (PBS, gray; LPS, teal). **i, j** Diagram of proposed WNT10B-FZD7 signaling pathway adapted from Ingenuity Pathway Analysis. **i** Shape colors indicate genes either upregulated (red; FPKM > 1, FDR < 0.05), present (gray; FPKM > 1, FDR < 0.05), or downregulated (light blue; FPKM > 1, FDR < 0.05) in BEC RNA-seq data after LPS. LPS (orange) binds to LBP, which stimulates the TLR4 pathway and an increase in WNT. **j** Red circle indicates the ligand WNT10B, upregulated after LPS in BEC RNA-seq data, and binds to the receptor FZD7 on endfeet. Blue shapes indicate proteins found in the endfoot proteome (uniquely identified or enriched ($p$ < 0.05) in Endfoot-TurboID compared to Endfoot-tdTomato samples). Yellow shapes indicate downstream genes also identified as upregulated by LPS in external astrocyte RNA-seq data[62] (FPKM > 1, FDR < 0.05). The data used to generate the schematics in Fig. 4 can be found in the Supplementary Fig. 10. Exact sample sizes per animal and source data are provided in a Source Data file.

upregulated in astrocytes at the RNA level, *Ccnd1* and *Cd44* (Fig. 6j, yellow shapes, and Supplementary Fig. 10c). Further, CD44 was found in our astrocyte and endfoot (PBS + LPS) proteome (Supplementary Data 1 and 5 and Fig. 4d). Taken together, our findings suggest that WNT10B-FZD7 signaling could be a means by which BECs translate peripheral events into the brain by detecting signals in the blood and directly communicating with endfeet to modulate astrocyte functions.

## BEC-endfoot ligand-receptor pairs are observed in human brain vessels and are altered in chronic neurological diseases

Immunofluorescence analyses identified the WNT10B-FZD7 pair in human vessels. To further investigate the level of conservation of other BEC-endfoot communication pathways between mouse and humans, we isolated the cerebral vasculature from adult human post-mortem frontal cortex samples (34–40 years old) and identified their proteome using LC-MS/MS (Fig. 7a, b, Supplementary Fig. 11a, and Supplementary Data 6). Comparison of the protein abundance of known brain vascular and parenchymal components between paired vessel and bulk cortex samples demonstrated the purity of our vascular preparation (Supplementary Fig. 11b–d). In addition, like with mouse tissue (Fig. 1g, k, l), we confirmed endfeet (AQP4 immunopositive) remained attached to the purified vessels (lectin; Fig. 7c, d). These data indicated that our human vascular preparation was of high purity and retained both BECs and endfeet so it could reliably be used to search for BEC-endfoot ligand-receptor pairs. We queried the human vascular proteome for the 59 endfoot receptors observed in mice (Fig. 2i) and found 44 (75%) of them (Fig. 7e). In addition, 194 of the 360 BEC-endfoot ligand-receptor pairs (54%) found in control mice, and 41 of the 86 differentially expressed after LPS (48%), were conserved between mouse and human (Fig. 7f–i).

To further explore the relevance of these ligand-receptor pairs to human physiology and pathology, we next examined their expression in published single nuclei RNA-seq (snRNA-seq) and brain-vasculature proteomics datasets of human cortex that compared multiple sclerosis (MS)[71] or Alzheimer's disease (AD)[72,73] samples to their respective age matched controls. The snRNA-seq data[71,73] allowed us to investigate expression of these ligand-receptor pairs with cellular resolution: ligands in the vasculature/endothelial cell clusters (pink, Fig. 8a, b) and the corresponding receptors in the astrocyte clusters (green, Fig. 8a, b). In addition, by including vasculature proteome data from human AD and control brains[72] we could investigate pair-expression alterations at the protein level (purple, Fig. 8b). We first looked for the presence of the ligand-receptor pairs we identified in mice in these datasets independently of disease-associated changes (Fig. 8c and Supplementary Data 7). Of the 371 pairs identified in mice, 309 (83%) were also found in one or more of these human datasets, including 41 of the 86 ligand-receptor pairs (47%) that were altered by LPS in mice

(Fig. 8c). A total of 34 ligand-receptor pairs were shared between all datasets, and 16 more were shared among all except for our human proteome. Within them, we found many recurrent pairs with receptors from the integrin, ephrin, plexin, and ERBB families, in addition to NOTCH2 and NRP2.

Out of all the ligand-receptor pairs shared between mouse and human, 55 were differentially expressed in human disease (Fig. 8d–f and Supplementary Data 7). The biggest overlap was found with MS, with fewer pairs significantly altered in AD, probably due to the lower number of differentially expressed genes in the AD reference dataset (Fig. 8e, f and Supplementary Data 7). Just 18 of these pairs were also altered by LPS in mice, with 13 changing in the same direction as in human disease (Fig. 8f teal bars, Supplementary Fig. 12, and Supplementary Data 7), indicating that many of the BBB responses to acute inflammation we identified in mice are different from human chronic diseases. Among the pairs differentially expressed in human disease, ITGAV receptor appeared to have a prominent role in MS as both the receptor and some of its ligands were upregulated in disease (ADAM15-, CCN1-, VWF-, FGF2-ITGAV; Supplementary Fig. 12 and Supplementary Data 7). Furthermore, semaphorin-NRP2 signaling stood out as a compelling candidate for further research. NRP2 was differentially expressed in both MS and AD but in opposite directions. In MS astrocytes, NRP2 was upregulated (RNA $\log_2$(MS/control) = 3.083, FDR < 0.001) with its ligand SEMA3F being also upregulated in BECs in both MS and LPS-injected mice (Supplementary Fig. 12 and Supplementary Data 7). SEMA3F is a modulator of vessel growth that competes with VEGF for NRP2 binding[74,75]. Although it has been shown to be expressed in actively demyelinating MS lesions, where it can influence remyelination[76], it is unclear which effects endothelial-derived SEMA3F could have by binding astrocytic NRP2. Other semaphorins have been directly linked with MS[77] and the BBB: for instance, blocking endothelial SEMA4D signaling can rescue BBB impairment in a mouse model of experimental autoimmune encephalomyelitis[78]. Conversely, in AD, NRP2 was downregulated in astrocytes (protein $\log_2$(AD/control) = −0.302, FDR = 0.002) and its ligands SEMA3B and SEMA3C were downregulated in LPS-injected mice (Supplementary Data 7). Additionally, *SEMA3F* gene locus has been associated with AD risk[79], and loss of endothelial semaphorin signaling through NRP2 has been shown to impair neuronal plasticity in the hippocampus and induce memory deficits in mice, implying this pathway may have an important role in AD pathogenesis[80].

Overall, these data suggest that mouse BEC-endfoot communication pathways across the BBB are largely conserved in the human brain and highlight its translational applicability by identifying ligand-receptor pairs in the human vasculature that are relevant to health as well as disease.

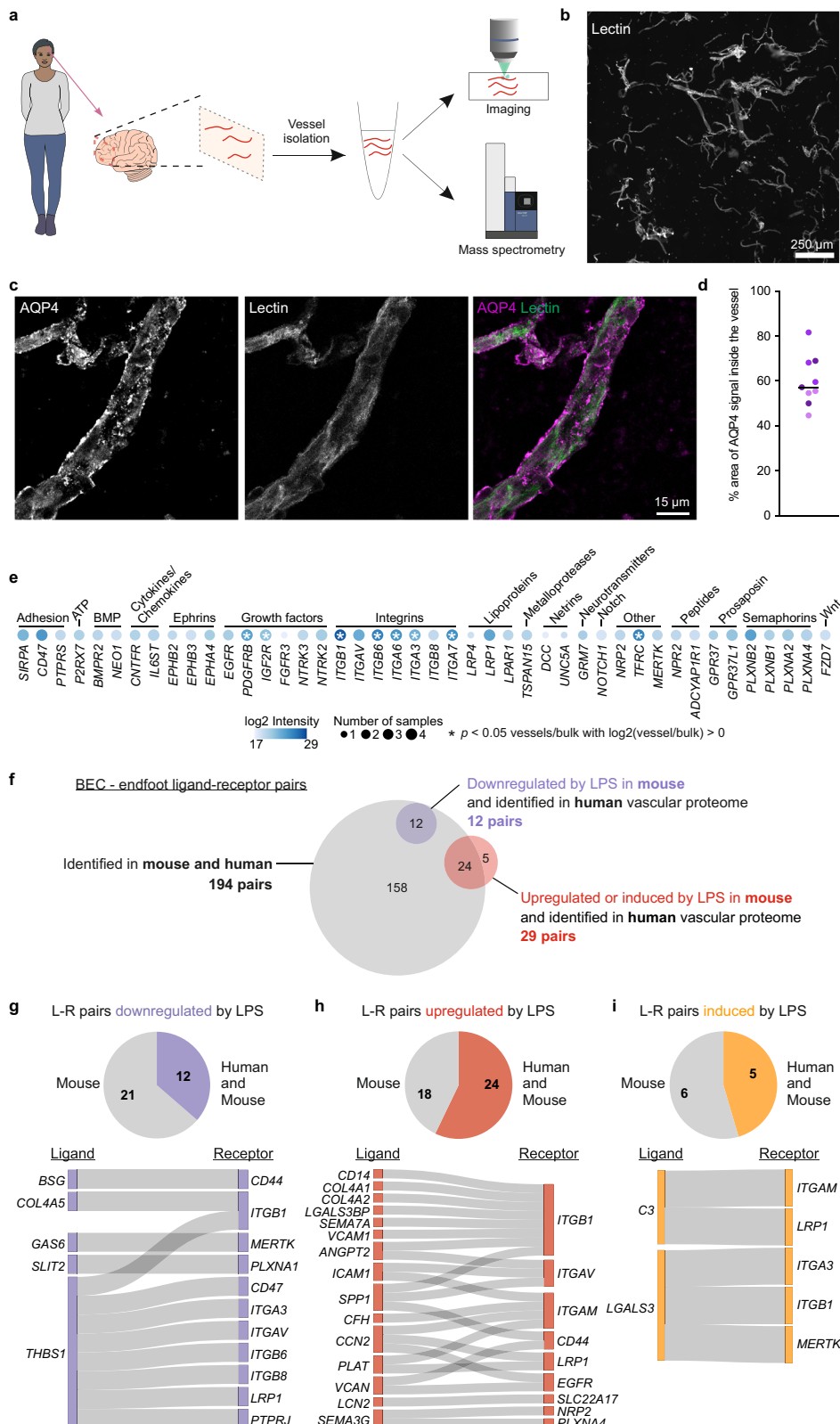

## Discussion

In search of the mechanisms involved in periphery-brain communication, we focused on the BBB and aimed to identify direct communication pathways between BECs and astrocyte endfeet (Fig. 1a). With this goal, we developed a method to tag and purify proteins from in vivo mouse endfeet (Figs. 1 and 2) that overcomes the limitations of previous attempts and identifies more endfoot proteins to further understand

the roles of this subcellular compartment in mouse and human brain physiology (Fig. 2 and Supplementary Figs. 2–5). Combining this method with BEC RNA-seq, with and without a peritoneal injection of LPS (Fig. 3 and Supplementary Figs. 6, 7), we identified hundreds of BEC to endfoot signaling pathways. Among those altered after LPS injection (Fig. 4), we characterized the WNT10B-FZD7 pair as an illustrative example of how signals from the blood can be translated to the brain via

**Fig. 7 | Conservation of mouse BEC-endfoot ligand-receptor pairs in human cortical vessel proteome. a** Schematic of human vessel isolation experiment for imaging and mass spectrometry. **b** Image showing a portion of isolated human microvessels stained with lectin. Scale bar = 250 μm. **c** Image of an isolated blood vessel stained for endfoot (AQP4, magenta) around the vessel (lectin, green). **d** Quantification of the percent area of the blood vessel, labeled with lectin, that colocalizes with the AQP4 signal. $N$ = 3 cortical brain samples with 3 vessel fragments analyzed per sample (see Table 1 in the Method's section for further information about the samples). Horizontal line represents the median. Scale bar = 15 μm. **e** Dot plots showing receptors identified in mouse endfoot proteomics (Fig. 2i) that are also found in the human vascular proteome. Intensity of each color corresponds to abundance of the protein, while the dot size corresponds to the

number of samples in which the protein was identified (total $N$ = 4 cortical brain samples). Asterisks (*) indicate proteins significantly enriched in the vessel proteome with a $\log_2$ (vessel/bulk) > 0 and $p$ < 0.05. **f** Venn diagram showing overlap of identified ligand-receptor pairs in mice at baseline (PBS, gray), upregulated by LPS (red), or downregulated by LPS (purple) that were also found in the human vascular proteome. **g–i** (Top) Pie charts illustrating the proportion of ligand-receptor pairs identified in mice that were also found in the human vascular proteome. (Bottom) Sankey plots illustrating interactions between shared ligand-receptor pairs that were downregulated (**g**), upregulated (**h**), or induced by LPS in mice (i.e., only present in the LPS condition; **i**). Proteins are referred to by their gene names. Exact sample sizes per animal and source data are provided in a Source Data file.

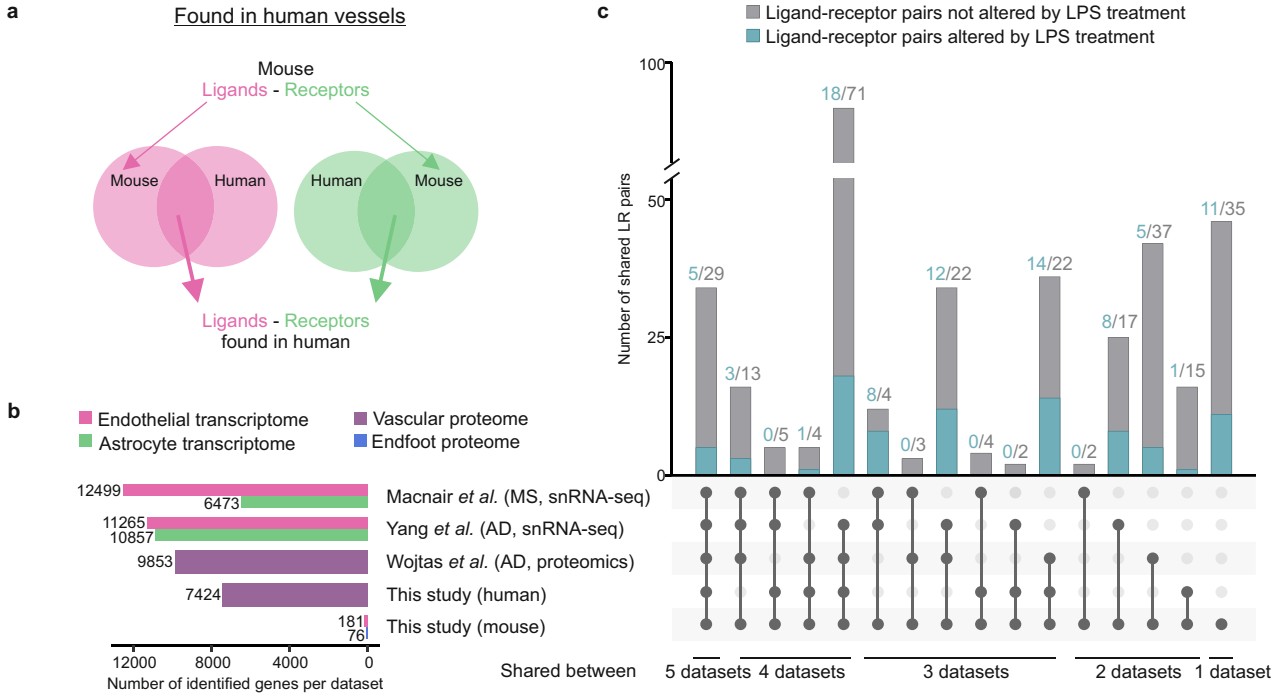

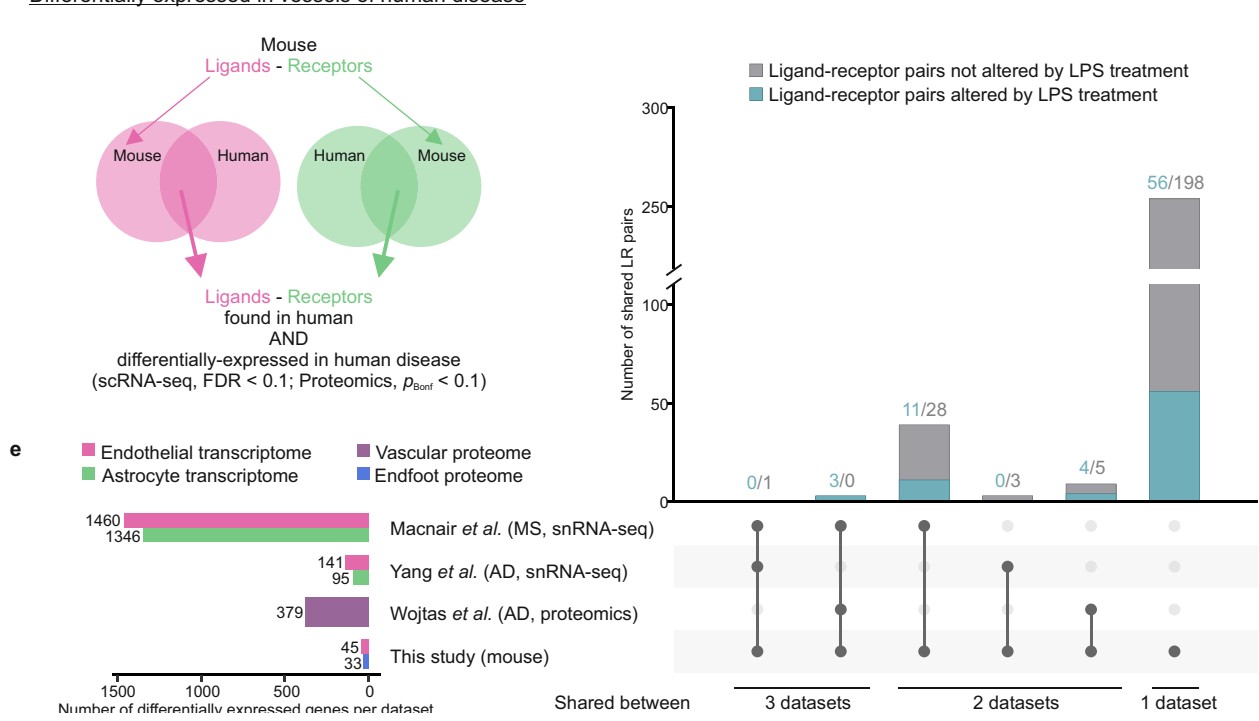

**Fig. 8 | Identification of ligand-receptor pairs between BECs and astrocyte endfeet that are relevant to human disease. a** Schematic of analysis. Three external human studies[71-73] and data from healthy human vascular proteome in this study (Fig. 7) were searched for the ligand-receptor pairs identified in mice (Fig. 4). **b** Bar charts indicating size of reference datasets. For snRNA-seq studies, BEC (pink) and astrocyte (green) clusters were considered; for proteomic studies, the entire vascular proteome (purple) was considered (see Supplementary Data 7). Bottom bars show the number of unique ligands in mouse BEC RNA-seq (pink) and number of unique receptors in mouse endfoot proteome (blue) identified in this study. **c** Number of mouse BEC-endfoot ligand-receptor pairs identified across all, some or none of the human datasets. Pairs found in multiple datasets indicated by connected dots. Number of ligand-receptor pairs indicated above bars (altered by LPS, teal; not altered, gray; number of pairs altered by LPS/number of pairs not altered). **d** Schematic to identify ligand-receptor pairs differentially expressed in human disease. Ligand-receptor pairs were considered differentially expressed in disease if either the ligand or receptor was differentially expressed in the reference dataset and its corresponding partner was identified. Transcripts in the reference datasets with a false discovery rate (FDR) less than 0.1 (disease vs. control) for snRNA-seq studies and proteins with a Bonferroni-adjusted $p < 0.1$ for proteomic studies were considered significant. **e** Bar charts indicating total number of differentially expressed genes in each of the referenced datasets. For snRNA-seq studies, endothelial cell (pink) and astrocyte (green) clusters were considered; for proteomic studies, the entire vascular proteome (purple) was considered. Number of unique ligands from mouse BEC RNA-seq (pink) and number of unique receptors from mouse endfoot proteome (blue) that contribute to differentially expressed ligand-receptor pairs are highlighted. **f** Number of differentially expressed ligand-receptor pairs identified across all, some or none or the referenced human datasets. Pairs differentially expressed in multiple datasets are indicated by connected dots. Number of ligand-receptor pairs indicated above bars (altered by LPS, teal; not altered, gray; number of pairs altered by LPS/number of pairs not altered). MS multiple sclerosis, AD Alzheimer's disease. Source data are provided in Supplementary Data 7.

a BEC-endfoot communication axis (Figs. 4–6 and Supplementary Figs. 8–10). Finally, we investigated the conservation of these pathways between mouse and human and identified many ligand-receptor pairs that were shared between the two with some that were affected in human neurodegenerative diseases (Figs. 7 and 8 and Supplementary Figs. 11, 12). Overall, our study defines the molecular underpinnings of BEC to endfoot communication in vivo via a wide variety of signaling pathways, demonstrates that BEC-endfoot interactions are dynamic and adaptable to events that occur in the periphery, and suggests these interactions are conserved between mouse and human.

The difficulty of investigating a subcellular compartment of a cell as complex as the astrocyte has hindered a comprehensive understanding of the endfoot's molecular machinery and breadth of functions. Endfoot RNA-seq has been used to identify the mRNAs that are locally translated in mouse endfeet[52]. However, not all endfoot proteins may be locally translated, and the relationship between the mRNA and protein levels is not always linear[81-83]. LC-MS/MS has been used to identify the proteome resulting from biochemical purification of astrocyte endfeet[37,39] or the interactors of known endfoot proteins i.e. hepacam[36] and AQP4[38] which, although enriched in the endfoot, are also found in other parts of the astrocyte. Notably, although these proteomes were mostly spare of non-endfoot markers (with the exception of Kameyama et al.[37]), they failed to identify all 10 of the most well-known endfoot proteins (Supplementary Fig. 3c). Comparison of our method to others (Supplementary Fig. 3) demonstrated a significant overlap with previous datasets and the discovery of over 800 additional proteins, including the 10 known endfoot proteins that were all enriched by either comparing the endfoot proteins to the tdTomato control (proteins present in endfeet and the denominated endfoot proteome in this study) or the whole astrocyte (proteins enriched in endfeet vs the rest of the astrocyte; Supplementary Data 2 and 3). Altogether, these results show that our method overcomes previous limitations and provides a more comprehensive description of the endfoot proteome.

Acute peripheral infections are a risk factor for neurodegenerative diseases and have severe effects on cognition[33-35,84,85]. However, the mechanisms by which these infections induce cognitive dysfunction are not well understood. Although often overlooked in neuroscience, BECs have been shown to undergo a strong response to LPS stimulation, with many inflammatory pathways upregulated[62,86,87] in vivo, and the conditioned media of LPS-stimulated BECs inducing astrocyte reactivity in vitro[29]. Indeed, we confirmed that BECs undergo dramatic gene expression changes in LPS injected mice. Our data suggest this can be driven by direct activation of BECs by LPS-TLR4 receptor binding (Fig. 6 and Supplementary Fig. 10). *Tlr4* expression by BECs has been shown in previous studies[88-90] with evidence for upregulation of the receptor and pathway in several mouse brain pathologies[91]. Accordingly, research on endothelial-specific *Tlr4* knock-out mice

demonstrate the importance of its endothelial signaling pathway for the health of the nervous system[92,93]. Nonetheless, it is also plausible that circulating cytokines or even neuroinflammatory responses of parenchymal cells also modulate BEC gene expression. We detected differential expression of genes encoding proteins released by BECs, including an abundant upregulation of inflammatory signals and downregulation of growth factors and extracellular matrix components (Fig. 3d and Supplementary Fig. 6e). These results serve as a reminder to consider BEC roles in the brain's inflammatory response in coordination with the other main players – microglia and astrocytes.

Among all pathways, WNT10B-FZD7 stood out due to the importance of WNT signaling for vascular development[65], the lack of previous data on BEC to astrocyte WNT signaling, and the association of both WNT10B and FZD7 with stroke[66,67]. Moreover, we detected the presence of this BEC-endfoot ligand-receptor pair in both mouse and human brain vessels (Figs. 5 and 6 and Supplementary Figs. 8, 9). WNT10B-FZD7 interactions have been described in non-neural cells[94], with a suggestion of both canonical and non-canonical FZD7 downstream pathways being described[95,96]. Interestingly, WNT10B expression can be activated via NF-κB[69,97,98] and our data suggests BECs detect LPS and subsequently upregulate NF-κB expression (Fig. 6 and Supplementary Fig. 10). Furthermore, we found that the vast majority of the WNT canonical pathway components are present in the endfoot, and astrocyte RNA-seq data from the same LPS model[62,99] detected activation of catenin beta-1 target genes (Fig. 6 and Supplementary Fig. 10). Future studies will be necessary to determine the direct relationship between these events and to explore the role of the many other ligand-receptor pairs we identified.

In vivo[30,31] and in vitro[27,28,100] data suggest BECs can directly signal to astrocytes. Our data corroborated the existence of previously described signaling pathways, *Bsg-Itga6*[31] and *Dll1/Dll4-Notch1*[27]. The combination of BEC-specific RNA-seq and endfoot proteomics allowed us to identify ligand-receptor pairs between BECs and endfeet with high confidence. Unlike solely RNA-based ligand-receptor pair analyses[31], the use of subcellular-compartment specific proteomics enabled us to focus on receptors that are, with high certainty, in the endfeet, as opposed to other areas of the astrocyte. This strategy provided a means to: (1) discover that BEC to endfoot signaling is ample and dynamic, and depends on peripheral events like infections that can modulate existing BEC-endfoot signaling pathways, but also induce others (Figs. 3 and 4), and (2) identify hundreds of BEC-endfoot ligand-receptor pairs that are present in the same cells of human post-mortem brains, with some of them being altered in neurodegenerative conditions, such as semaphorins-NRP2 (Figs. 7 and 8). Our research opens up many opportunities to use these tools in new contexts, and to explore the implications of the pathways we identify here for conserved mechanisms between mouse and human that lead to inflammation-associated cognitive dysfunction[33,34,84,85].

Several considerations will need to be made when interpreting our datasets. (1) We observed a small fraction (1.3%) of CC1 positive cells that colocalized with streptavidin signal (Supplementary Fig. 1d, e) and *Mog* in the endfoot proteome (Supplementary Figs. 2a and 3b). However, there was no significant expression of other oligodendrocyte markers like *Sox10*, *Mobp*, or *Gjc2* (Supplementary Figs. 2a and 3b), suggesting that, if there was contamination of oligodendrocyte proteins in the endfoot proteome, this would be minimal. (2) We analyzed our TurboID vs tdTomato proteome datasets using the workflow presented in Fig. 2b–g. Non-biotinylated proteins can bind non-specifically to the streptavidin-coated magnetic beads (Fig. 2c). Therefore, by concentrating on proteins that are specifically enriched in TurboID samples relative to tdTomato controls, we can confidently identify proteins that are specifically biotinylated in astrocytes or endfeet. Moreover, the filtering parameters for protein identification can be adjusted *ad nauseam*, such as varying the thresholds for *p*-value or the minimum number of samples in which a protein should appear to consider it as a true positive. For example, we found that when making comparisons between proteomes that are more similar between each other, e.g. endfoot-TurboID LPS vs PBS as opposed to TurboID vs tdTomato, using a more stringent minimum sample threshold reduces variability in the data and provides more robust results. (3) The data in Fig. 3e, f shows early signs of proteome remodeling. Further analyses at later timepoints may reveal the subsequent consequences of such remodeling. (4) Our experimental workflow and analyses have the capacity to detect intercellular (BEC-endfoot) communication pathways but do not exclude the possibility that some of the ligand-receptor pairs identified also mediate within-cell signaling. For example, in mouse, we observed that FZD7 protein expression was endfoot specific when compared to vascular cells (Fig. 5), but in humans, there was clear expression of FZD7 in both endfeet and vascular cells (Supplementary Fig. 9a), suggesting that within-cell communication through the WNT10B-FZD7 pathway might be an adaptive difference between mice and humans. (5) The human data included in this manuscript originates from healthy and chronic neurodegenerative disease samples. Thus, its comparison to the mouse ligand-receptor pairs that change after LPS injection is limited by the lack of data from individuals with an acute inflammatory challenge.

In conclusion, our study paves the way for future exciting research to understand the contributions of BEC-endfoot interactions to brain health and disease and features the role of BECs in the brain's inflammatory response in coordination with neural cells. Acute peripheral infections are a risk factor for neurodegenerative diseases and have severe effects on cognition[33–35], but the mechanistic relationship between infections and cognitive dysfunction are not well understood. The observation of altered expression of BEC-endfoot communication pathways in human chronic neurodegenerative disease highlights the immediate importance of assessing the drivers of these alterations and their contribution to conditions that lead to cognitive decline.

## Methods

### Animals
All experiments were conducted under the UK Home Office Animals (Scientific Procedures) Act 1986, in agreement with local ethical and veterinary approval (Biomedical Research Resources, University of Edinburgh), and authorized by Project Licenses PP8310627 and PP2636156. Most experiments were performed using C57Bl/6J mice (Charles River Laboratories, UK) with male and female mice in equal proportion. For brain endothelial cell RNA-seq experiments *Cdh5-Cre/ERT2* (Tg(Cdh5-cre/ERT2)1Rha)[64] mice were bred with B6N.129-*Rpl22*$^{tm1.1Psam}$/J (RRID: IMSR_JAX: 011029)[63]. Mice were housed in techniplast GM 500 cages with a rodent roll, dome home, chew stick and handling tube, in a 12-h light/dark cycle with food and water *ad libitum*,

at 20–24 degrees Celsius and with humidity between 45-65%. Mice used were between 12 and 22 weeks of age.

### Drug administration
Lipopolysaccharide (LPS) (Sigma Cat #L5024) from *Escherichia coli* O127:B8 was dissolved in sterile PBS at a concentration of 1 mg/ml and stored in aliquots at −80 °C. LPS-treated mice received one single intraperitoneal (i.p.) injection of 5 mg LPS/kg body weight. Control mice received an equivalent volume of sterile phosphate buffer saline (PBS). All mice used for proteomics in this manuscript received either LPS or PBS. Mice were weighed before and 24 h after the injection. For inducible gene expression in CRE-dependent mouse lines, tamoxifen (Sigma Cat #T5648) was dissolved in sunflower oil at a concentration of 20 mg/ml and was administrated at 100 mg tamoxifen/kg body weight by oral gavage for 5 consecutive days.

### Generation of adeno-associated viruses
The pUCmini-iCAP-PHP.eb plasmid used for the adeno-associated AAV capsids was kindly shared by Dr. Viviana Gradinaru (Addgene Cat# 103005). The pZac2.1-GfaABC$_1$D-tdTomato plasmid used as the control AAV was kindly shared by Dr. Baljit S. Khakh (Addgene Cat# 44332). pZac2.1-GfaABC1D-GFP was generated by excising GFP with BamHI from pZac2.1-GfaABC1D-AQP4-GFP, a gift from Dr. Baljit S. Khakh, and amplifying with the following primers before inserting into the BamHI site of the pZac2.1-GfaABC1D vector using the In-fusion® HD Cloning kit (Takara Cat# 639650): Forward: 5′ CCTCGAGCTCGGATCCGCCAC-CATGGTGAGCAAGGGCGAGGAG 3′; Reverse: 5′ TAAGCGAATTG-GATCCTTACTTGTACAGCTCGTCCAT 3′. CMV-V5-TurboID-NES-pCDNA3 was kindly shared by Dr. Alice Ting (Addgene Cat#107169) and we cloned the cDNA of the TurboID enzyme gene into a pZac2.1 plasmid containing the astrocyte specific promoter GfaABC$_1$D. To do so, the sequence containing the V5 Tag-TurboID-HA Tag-nuclear export signal (NES) was amplified by PCR using the following primers: Forward: 5′ CCTCGAGCTCGGATCCATGGGCAAGCCCATCCCCAA 3′; Reverse: 5′ TAAGCGAATTGGATCCTTAGTCCAGGGTCAGGCGCTCCAGGGG 3′. The vector GfaABC$_1$D-BioID2-13xLinker-BioID2, kindly shared by Dr. Baljit S. Khakh, was cut with the restriction enzyme *BamHI* to remove the BioID2 gene. The V5-TurboID-NES-HA fragment was then inserted into the *BamHI* pZac2.1-GfaABC$_1$D vector using the In-fusion® HD Cloning kit (Takara Cat# 639650). Plasmids were confirmed by sequencing (Source Bioscience).

In preparation for AAV packaging, HEK 293T cells ((ATCC)293T/17 CRL-11268) were cultured at 37 °C, 5% $CO_2$ in Dulbecco's Modified Eagle Medium (DMEM, Gibco Cat# 11965092) with 10% heat-inactivated fetal bovine serum (FBS; Gibco Cat# A5256801). Cells were cultured in 145 mm$^2$ plates (Greiner Bio Cat# 639160) at 80% confluence on the day of transfection. Plasmids pZac2.1-GfaABC1D-TurboID or pZac2.1-GfaABC1D-tdTomato were co-transfected into HEK 293T cells with the pALD-X80, AAV helper Plasmid (Aldevron, Cat# 5017-10), and pUCmini-iCAP-PHP.eb packaging plasmid at the molecular ratio of 1:3:1. Viral particles were harvested at 72 h post-transfection and purified by POROS™ CaptureSelect™ AAVX Affinity Resin (Thermo Fisher, Cat# A36741). The purified rAAVs were titered by ddPCR using the ITR primers (BioRad Cat# QX200, Forward primer: 5′ GGAACCCCTAGTGATGGAGTT 3′; Reverse primer: CGGCCTCAGT-GAGCGA 3′). The viral vectors were aliquoted and stored at −80 °C until use.

### AAV administration and in vivo TurboID protein biotinylation
Mice received a single intravenous injection in the tail vein of either PHP.eb-GfaABC$_1$D.V5-TurboID-HA-WPRE or PHP.eb-GfaABC$_1$D-tdTomato (1.5 ×10$^{11}$ genome copies) and given at least three weeks for the virus to express. Biotin (Sigma Cat# B4501) was dissolved in PBS at 5 mM, and mice were given a series of three biotin subcutaneous injections (24 mg

biotin/kg body weight each) at 24, 18, and 3 h prior to dissection. The first biotin injection (24 h before dissection) occurred immediately following PBS/LPS treatment.

For assessment of FZD7 signal within endfeet, mice received PHP.eb-GfaABC$_1$D-GFP AAVs (1.5 ×10$^{11}$ genome copies) to tag astrocytes with the fluorescent reporter GFP and given at least three weeks for the virus to express.

## Immunofluorescence

For mouse brain immunofluorescence studies, mice were given a terminal dose of anesthesia (Dolethal, 200 mg/mL, 0.1 mL/animal) and transcardially perfused with PBS containing 10 units/mL of heparin followed by 15 mL 10% formalin. Brains were dissected, stored in 10% formalin at 4 °C overnight, and moved to 30% sucrose until sectioning. Coronal or sagittal sections (40 µm each) were cut using a cryostat (Leica) and kept in antifreezing solution (0.05 M PBS, 250 mM sucrose, 7 mM MgCl$_2$ and 50% glycerol) at −20 °C.

Mouse brain sections were washed twice in PBS for 10 min and incubated at room temperature for 2 h in blocking solution (10% normal goat (Stratech, Cat# 005-000-121-JIR) or donkey (Stratech, Cat# 017-000-121-JIR) serum in PBS with 0.2% Triton X-100 (Sigma, Cat# 102533092) with agitation. Sections were subsequently incubated overnight in primary antibodies diluted in the blocking solution at 4 °C. The following primary antibodies were used: rabbit anti-S100β (1:1000, Abcam Cat# ab41548), mouse anti-S100β (1:1000, Sigma Cat# S2532), mouse anti-NEUN (1:500, Sigma Cat# MAB377), guinea pig anti-S100β (1:500, Synaptic Systems Cat# 287004), rabbit anti-NEUN (1:2000, Cell Signaling Cat# 12943S), rabbit anti-RFP (1:1000, Rockland Cat# 600-401-379), goat anti-IBA1 (1:1000, Abcam Cat# ab5076), rabbit anti-IBA1 (1:1000, Wako Cat# 019-19741), rabbit anti-aquaporin4 (1:1000, Millipore Cat# ab3594), mouse anti-β-dystroglycan (1:1000, DSHB Cat# 7D11), rabbit anti-HA (1:500, Cell Signaling, Cat# C29F4), rat anti-PECAM1 (1:50, BD Biosciences Cat# 550274), goat anti-CD13 (1:100, R&D Cat# AF2335), mouse anti-CC1 (1:300, Sigma Cat# OP80), rabbit anti-FZD7 (1:500, Abcam Cat# ab64636), rabbit anti-WNT10B (1:200, Abcam Cat# ab70816), biotinylated lectin (1:500, Vector Cat# B-1175-1), and rabbit anti-PYGB (1:500, Atlas antibodies Cat# HPA031067). After overnight incubation at 4 °C with primary antibodies, sections were washed three times in PBS for 10 min each and incubated with the secondary antibody diluted in blocking solution. The following secondary antibodies were used at a concentration of 1:1000 unless otherwise indicated: goat anti-mouse IgG Alexa 488 (Invitrogen Cat# A11001), goat anti-rabbit IgG Alexa 488 (Invitrogen Cat# A11008), donkey anti-mouse IgG Alexa 488 (Invitrogen Cat# A21202), donkey anti-rabbit IgG Alexa 488 (Invitrogen Cat# A21206), goat anti-rabbit IgG Alexa 546 (Invitrogen Cat# A11010), donkey anti-rabbit IgG Alexa 546 (Invitrogen, Cat# A10040), donkey anti-rat IgG Cy3 (Jackson Immuno Research Cat# 712-165-150), goat anti-mouse IgG Alexa 647 (Invitrogen Cat# A21235), goat anti-rabbit IgG Alexa 647 (Invitrogen Cat# A21244), goat anti-rat IgG Alexa 647 (Invitrogen Cat# A21247), donkey anti-goat IgG Alexa 647 (Invitrogen Cat# A21447), and conjugated streptavidin Alexa 405 (Invitrogen, Cat# S32351). For rabbit anti-WNT10B staining, antigen retrieval was performed by incubating tissue in methanol at −20 °C for 30 min. To visualize biotinylated proteins, brain sections were incubated with streptavidin Alexa 488 (1:200, Invitrogen Cat# S32354). Nuclei were identified with either Hoechst 33342 (ThermoFisher, Cat# H3570) or DAPI (Sigma, Cat# D9542). The sections were mounted on microscope slides in ProLong Gold antifade reagent (Invitrogen, Cat# P36930) and coverslipped (170 ± 5 µm, Marienfeld, Cat# 0107222).

For immunofluorescence of expanded tissue, CUBIC-X expansion microscopy and tissue clearing[68] was performed on 200 µm thick brain slices from wild-type mice that had received PHP.eb-GfaABC1D-gGFP AAVs to tag astrocytes with the fluorescent reporter eGFP. CUBIC-X reagents were provided by TCI Chemicals and the 'CUBIC – Animal Tissue-Clearing Reagents - Technical Guidebook' available on the TCI Chemicals website was used to develop our protocol. Free-floating sections were initially washed with 1X PBS (3 × 10 min each) to remove anti-freezing media. CUBIC-L (TCI Chemicals, Cat# T3740) was diluted 1:1 with distilled water to prepare a 50% working solution. The slices were immersed in 50% CUBIC-L solution and incubated at 37 °C for 12 h with gentle agitation. After 12 h, the CUBIC-L solution was removed and the slices were washed with 0.5% Triton-X diluted in 1X PBS at room temperature with gentle agitation 3 × 10 minutes each, followed by immersion in CUBIC-X1 solution (TCI Chemicals, Cat# T3866) and incubated at 4 °C for 24 h with gentle agitation. Our normal immuno-fluorescence protocol was used for protein detection, and the following primary and secondary antibodies were used: primaries chicken anti-GFP (1:1000, Abcam, Cat# ab13970), rabbit anti-FZD7 (1:500, Abcam, Cat# ab64636) and biotinylated-lectin (1:200, Vector, Cat# B-1175-1); secondaries goat anti-Rabbit Alexa 647 (1:1000, Invitrogen, Cat# A21244), goat anti-chicken Alexa 488 (1:1000, Invitrogen, Cat# A11039) and Streptavidin Alexa 405 (1:200, Invitrogen, Cat#S32351). After immunostaining, the slices were immersed in CUBIC-X2 solution (TCI Chemicals, Cat# T3867) and incubated for 24 h at room temperature with gentle agitation. The slices were then mounted onto SuperFrost Plus glass slides with parafilm chambers to confine the tissue without compression (mounting medium TCI Chemicals, Cat# M3292).

For immunofluorescence of isolated microvessels, the area containing the vessels was outlined with a PAP pen (Abcam, ab2601) to create a hydrophobic barrier, and all washes and incubation steps were performed directly on the slide. Mouse and human isolated vessels were visualized by incubating with fluorescein-labeled lectin (1:200 for mouse vessels, 1:1000 for human vessels, Vector Cat# 1171), and/or biotinylated lectin (1:200, Vector Cat#B-1175), the endfeet with rabbit anti-aquaporin4 (1:1000, Millipore Cat# ab3594) and smooth muscle cells with mouse anti-α-Smooth Muscle Actin (ACTA2) (1: 200, Sigma-Aldrich Cat#A5228). For FZD7 staining, antigen retrieval was performed by incubating the vessels in 0.01 M sodium citrate, pH = 6, at 97.5 °C for 20 min. Mouse isolated vessels were visualized by incubating with fluorescein-labeled lectin 1:200 (Vector Cat# 1171) or biotinylated lectin (Vector Cat# B-1175-1), the endfeet with mouse anti-aquaporin4 (1:100, Abcam, Cat# ab9512), and rabbit anti-Frizzled7 (1:250, Proteintech, 16974-1-AP). The following secondary antibodies were used at a concentration of 1:1000: goat anti-mouse 546 (Invitrogen, Cat# A11003), goat anti-rabbit IgG Alexa 647 (Invitrogen, Cat# A21244), and Streptavidin Alexa 405 conjugate (1:200, Invitrogen, #S32351) to visualize biotinylated lectin. The sections were mounted on microscope slides in ProLong Gold antifade reagent (Invitrogen, Cat# P36930) and coverslipped (170 ± 5 µm, Marienfeld, Cat# 0107222).

Human tissue for vessel purification and staining was sourced from Edinburgh Brain and Tissue Bank (Table 1). Informed consent was either obtained from participants in life and/or from the nearest relative for deceased participants, in keeping with legal requirements. Sex information is based on clinical data.

For immunofluorescence studies in human brain slices, formalin-fixed paraffin-embedded (FFPE) frontal cortex (BA46) human brain sections (8 µm) were obtained from the Edinburgh Brain Bank, a Medical Research Council funded facility with research ethics committee (REC) under the ethical approval (21/ES/0087). For immuno-fluorescence labeling, sections were first deparaffinized with two 5 min xylene washes and rehydrated in consecutive 5 min washes in 99.8% and 70% ethanol. Sections were rinsed in distilled water for 5 min prior to performing antigen retrieval in 0.01 M sodium citrate (pH 6.0) at 97.5 °C in a water bath for 30 min. Sections were cooled in PBS for 10 min. A hydrophobic border was applied using a PAP pen (Merck, Cat # Z672548) and sections were incubated for 1 h in blocking solution

**Table 1 | Tissue used for immunostaining**

| BBN ID | PMI | Brain area | Known clinical diagnoses | Cause of death | Tissue type |
|---|---|---|---|---|---|
| BBN001.30972 | 99 | BA10 | No known abnormality | Ischemic heart disease, coronary artery atherosclerosis | Purified vessel |
| BBN001.30841 | 103 | BA10 | No known abnormality | Myocardial infarction; Coronary artery atherosclerosis | Purified vessel |
| BBN_21396 | 36 | BA6/8 | Depressive episode, unspecified; Essential (primary) hypertension; Type 2 diabetes mellitus; Pure hypercholesterolemia | Ischemic heart disease; Coronary artery atherosclerosis; Type 2 Diabetes Mellitus | Purified vessel |
| BBN001.28959 | 86 | BA46 | Sciatica. Low back pain. Other symptoms and signs involving emotional state. Other stressful life events affecting family and household. | Ischemic heart disease Coronary artery atherosclerosis | Brain slide |
| BBN001.26124 | 89 | BA46 | No abnormality detected | Haemopericardium Ruptured acute myocardial infarction Coronary artery thrombosis Coronary artery atherosclerosis | Brain slide |
| BBN_22618 | 40 | BA46 | Depressive episode. Acute and transient psychotic disorders. Mental and behavioral disorders due to use of cannabinoids. | Suspension by ligature | Brain slide |

Human subjects were chosen based on the following criteria: age 20–45 years, postmortem interval (PMI) < 100 h, no co-occurring neuropathology (e.g. Lewy body, TDP-43) and no co-occurring neurodegenerative disease (e.g. Parkinson's disease, multiple sclerosis). For the experiments listed in this table four males and two females between 34 and 45 years old were included.

(5% normal donkey serum (Stratech, Cat# 017-000-121-JIR) in PBS with 0.3% Triton X-100 (Sigma, Cat# 102533092)). Subsequently, sections were incubated with primary antibodies diluted in antibody diluent solution (1% normal donkey serum in PBS with 0.3% TritonX-100). The following primary antibodies were used: rabbit anti-FZD7 (1:200, Abcam Cat# ab64636), rabbit anti-WNT10B (1:200, Abcam Cat# ab70816), rabbit anti-PYGB (1:200, Atlas antibodies Cat# HPA031067), mouse anti-ALDH1L1 (1:200, NeuroMab, Cat# 75-140), and goat anti-VE-cadherin (1:200, R&D Systems, Cat# AF938). Sections were washed three times in PBS for 10 min each and incubated with fluorophore-conjugated secondary antibodies in antibody diluent solution for 1 h at room temperature in the dark. The following secondary antibodies were used: donkey anti-rabbit IgG Alexa 488 (1:250, Invitrogen Cat# A21206), donkey anti-mouse IgG Alexa 546 (1:250, Invitrogen, Cat# A10036), donkey anti-goat IgG Alexa 647 (1:250, Invitrogen Cat# A21447). Following three washes in PBS, tissue autofluorescence was quenched by incubating sections in TrueBlack Lipofuscin Autofluorescence Quencher (1:20 in 70% ethanol, Biotium, Cat# 23007) for 30 s followed by five washes in PBS of 5 min each with gentle shaking. Nuclei were stained using DAPI (Sigma, Cat# D9542). Sections were coverslipped (170 ± 5 μm, Marienfeld) using ProLong Gold antifade reagent.

## RNAscope in situ hybridization

RNAscope (Advanced Cell Diagnostics; ACD) in situ hybridization was performed on 40 μM cortical mouse brain tissue sections. Free-floating sections were initially washed three times in 1X PBS for 10 min each to remove anti-freezing media and mounted onto SuperFrost Plus glass slides. Slides were left to air dry for at least 3 h at room temperature, then baked at 60 °C for 30 min, followed by incubation with pre-chilled 10% formalin diluted in 1X PBS for 25 min at 4 °C. RNAscope Multiplex Fluorescent V2 Assay (ACD Cat# 323100) was used to label multiplex probes. The assays were performed according to the manufacturer's instructions and user manual for fixed-frozen tissue, including slightly adjusting and following steps of tissue dehydration, pre-treatment with hydrogen peroxide and 1X Target Retrieval. This pre-treatment method was optimized for recognition of probes and combined immunofluorescence antibodies for both RNA and protein detection. For the 1X Target Retrieval steps, the assay was optimized by initially pre-heating the 1X Target Retrieval solution to 97.5 °C for 15 min before use. After hydrogen peroxide incubation (following the user manual), the slides were then immersed into the pre-heated 1X Target Retrieval solution for 15 min only and immediately washed with distilled water, followed by immersion into 100% ethanol for 5 min. The slides were then dried at 60 °C for 10 min and left overnight at room

temperature prior to protease treatment. RNAscope Protease III solution was used for protease treatment, following the user manual, and adjusting the incubation step to 15 min at 40 °C to limit over-digestion. RNAscope probe Mm-Wnt10b (ACD Cat# 401071) paired with TSA Vivid 650 (1:1500) fluorophores were used in this study. After developing with HRP and prior to the addition of blocking buffer following our immunofluorescence protocol, the slides were washed with PBS for 10 min. It is important to note that the slides were not left dry at any point during the immunofluorescence steps. Immunofluorescence was performed as described above, and the following primary and secondary antibodies were used: primary antibody rabbit anti-ERG (1:500, abcam, Cat# ab92513) and biotinylated-lectin (1:500, Vector, Cat# B-1175-1); secondary antibody goat anti-rabbit 488 (1:1000, Invitrogen, Cat# A11008) and streptavidin Alexa 405 (1:1000, Invitrogen, Cat# S32351).

## Microscopy and image analysis

Fluorescent images were taken on a Zeiss LSM900 confocal laser-scanning microscope.

To examine the *distribution of expression of astrocyte-TurboID and astrocyte-tdTomato* in the mouse brain (Supplementary Fig. 1a, f), a human-guided automatic epifluorescence tile scan was carried out. For the rest of images, confocal microscopy was used.

*For AAV cell-specificity validation* (Supplementary Fig. 1d, e, g, h), 3 images (1024 ×1024 pixels) were taken in the prefrontal cortex (PFC) area using a 25 × 0.8 NA objective (Plan-Apochromat, Zeiss Cat# 420852-9871-000). Image analysis was conducted using FIJI (ImageJ). Three images were taken in the PFC area for each cell marker (S100β, NeuN, Iba1, CC1). Z-stacks of 30 μm (30 slices, 1 μm step size) were taken. For each image, the total number of cells expressing the cell-specific marker, and the number of cells co-stained with streptavidin, was counted manually. The percent co-localization for each image was calculated by dividing the total number of double-labeled cells (i.e., labeled with streptavidin and the cell-specific marker) by the total number of cells expressing cell specific markers.

*For streptavidin intensity measurements* (Supplementary Fig. 1b, c), the laser power and gain were kept consistent for all images. To calculate the streptavidin signal intensity as a percentage of total area, a minimum threshold for streptavidin signal was calculated based on the background streptavidin signal present in brains expressing the tdTomato control AAV. Then, the signal intensity was captured through the "measurements" tool within FIJI. The average minimum threshold calculated was applied to all images from brains expressing TurboID AAV, and the area of streptavidin signal intensity above the threshold was captured and recorded.

*To measure endfoot-vasculature distance* (Fig. 1e), a 20 μm line was drawn from the center of the vessel in 63× (oil objective, 1.2 NA, Plan-Apochromat, Zeiss Cat# 420882-9870-799) confocal single planes and the intensity profile was plotted. Intensity values for each channel were exported from FIJI and analyzed separately with Clampfit (Molecular Devices) to determine the position of the highest intensity peak for each marker. This data was then used to calculate the distance between markers.

*The proportion of different sizes of vessels after vessel purification* (Fig. 1j) was quantified using FIJI. A ROI for all vessels in the image was generated using the lectin channel and the percent area was recorded. Next, vessels with a diameter less than 10 μm were selected and deleted from the image and the percent area was recorded again. This process was subsequently repeated for vessels less than 30 μm and greater than 30 μm. The vessel composition was calculated by subtracting the percent areas for each vessel size from the full lectin ROI in the order they were acquired and normalizing each area by the total percent area of the identified vessels.

*To examine the aquaporin-4 coverage in PBS and LPS isolated vessel samples* (Fig. 1j–n; Supplementary Fig. 7d–j), 1-7 images of each vessel type (i.e., artery, arteriole, capillary, vein and venule) were taken in each sample using a 25× objective. α-SMA signal was observed in all vessels >8–10 μm in diameter[53]. Thus, vessel type was determined by the combination of vessel diameter and α-SMA morphology (i.e., arteries and arterioles: 30–90 μm and 8–30 μm diameter respectively, with cross-vessel striated α-SMA signal; veins and venules: 30–90 μm and 8–30 μm diameter respectively, with irregular α-SMA signal along the vessel). Using FIJI, Z-projections of the lectin and aquaporin-4 channels were generated and a ROIs representing the vessel were created thresholding the lectin signal. AQP4 percentage area of colocalization was then measured within the vessel ROIs. In bigger vessels containing more parts of the vascular tree, a ROI of the contour of the vessel was created prior to measuring AQP4 coverage.

*The proportion of astrocyte endfeet obtained from isolated different sized vessels* (Fig. 1n) was calculated by first averaging the percent AQP4 coverage for each vessel size category (i.e., for 10–30 μm, the percent AQP4 coverage for arterioles and venules and for >30 μm, the percent AQP4 coverage for arteries and veins were averaged together). The average percent AQP4 area was then multiplied by the normalized proportion of each vessel size (in Fig. 1j) and divided by the total to obtain the proportion of endfeet coming from each vessel category in our sample.

*For PYGB analyses* (Supplementary Fig. 5), a β-dystroglycan and lectin signal intensity threshold were applied to generate two ROIs representing endfeet and vessels respectively in mouse, and ALDH1L1 or VE-cadherin to create ROIs for astrocyte endfeet or vessels in human samples. Then the PYGB percent area of colocalization was measured within the endfoot or vessel ROIs.

*For FZD7 analysis in expanded tissue* (Fig. 5a, b), CUBIC-X sections were imaged with a 63× oil objective using Airyscan mode and processing (Zeiss) on a Zeiss LSM900 confocal laser-scanning microscope. For image analysis with FIJI, a GFP or lectin signal intensity threshold was applied to generate a region of interest representing either the endfoot or vessel, respectively. FZD7 percentage area of colocalization was then measured within the endfoot or vessel ROIs.

*To measure FZD7, AQP4, and lectin intensity profile and distance in isolated vessels* (Fig. 5c–i) a 15 μm line was drawn across the vessel in 63× (oil objective, 1.2 NA, Plan-Apochromat, Zeiss Cat# 420882-9870-799) confocal single planes and the intensity profile was plotted as shown by representative examples in Fig. 5d, e. Intensity values and their respective distance across the drawn line for each channel were exported from FIJI and analyzed separately with Clampfit (Molecular Devices). The position in μm of the highest intensity peak for each marker was used to calculate the distance of FZD7 signal from AQP4 and lectin as well as the distance of AQP4 from lectin as shown in Fig. 5f

and plotted in Fig. 5g. In addition, for each marker the fluorescence intensity ratio was determined as the difference between the highest intensity peak and the baseline intensity in the middle of the vessel, as shown in Fig. 5h and plotted in Fig. 5i. Both, the distance between markers and the intensity ratio values that are shown in the figures are the result of the average value between the peaks of each side of a single the vessel.

*For RNAscope image analysis* (Supplementary Fig. 8), sections were imaged with a 63× objective using Airyscan mode on a Zeiss LSM900 confocal laser-scanning microscope and analyzed using FIJI. A lectin signal intensity threshold was applied to generate a region of interest representing whole astrocytes or vessels, respectively. Probe signal of *Wnt10b* within these respective ROIs was analyzed by recording the percent area of *Wnt10b* signal within the lectin signal, respectively. An additional threshold was applied to the *Wnt10b* signal to allow for the manual counting of each RNA dot found within the lectin ROIs.

*For WNT10B and FZD7 immunohistochemistry* quantification analyses in mice (Fig. 6), vessels <10 μm in diameter were imaged with a 63× objective using the Airyscan mode and processing (Zeiss) and analyzed using FIJI. For FZD7, a β-dystroglycan signal intensity threshold was applied to generate a region of interest (ROI) representing endfeet. FZD7 signal within this ROI was analyzed by recording the percent area and mean-intensity of FZD7 signal within the β-dystroglycan signal. In addition, we generated a histogram representing how many FZD7 positive pixels were found at each signal intensity value within the endfoot (defined by the β-dystroglycan signal) and recorded the intensity at which most of the FZD7 positive pixels are found (peak intensity). A similar process was followed for WNT10B, but due to WNT10B being a ligand that could be released by the BECs, the WNT10B signal was quantified within a ROI that contained both the astrocyte endfoot and the vessel by using β-dystroglycan staining to define the endfoot boundary.

*For FZD7 and WNT10B analyses in postmortem human brain* (Supplementary Fig. 9), fluorescent images were acquired with a 63× objective and analyzed using FIJI. To determine the percent area labeled by FZD7, an ALDH1L1 signal intensity threshold was applied to generate a ROI for astrocyte endfeet. To measure WNT10B, a vasculature ROI that encompassed both the vessel and the surrounding endfeet was first generated. The ROI was defined by VE-cadherin staining, which was expanded outward by thirty pixels to approximate the endfoot ALDH1L1 immunopositive boundary. Within this vessel ROI, the percent area labeled by WNT10B was calculated.

*For analysis of human isolated vessels* (Fig. 7), the overlap between aquaporin-4 (AQP4) and human isolated vessels of <10 μm in diameter was imaged with a 63× objective and analyzed using FIJI. A region of interest (ROI) representing the blood vessel was generated by applying a threshold to lectin signal intensity, and the overlap of AQP4 signal with the lectin ROI was recorded as percent area.

## Mouse brain microvasculature isolation

Isolation of brain microvessels was conducted by adapting a published protocol[53]. Briefly, mice were culled by cervical dislocation, and the brain was quickly removed and dissected in cold PBS. Isolated cortices were snap-frozen in liquid nitrogen and stored at −80 °C until microvessel isolation. For each mouse, the cortex was divided into two hemispheres: one hemisphere was used for microvessel isolation and endfoot proteomics, and the other hemisphere was used for whole astrocyte proteomics.

For isolation of brain microvessels, one cortex hemisphere was homogenized in 2 mL of MCDB-131 media (Gibco Cat# 10372019) with Halt protease inhibitor (ThermoFisher Cat# 78430) using a dounce tissue grinder (Kimble Cat# 885303-002). The homogenate was centrifuged (2000 × *g*/5 min/4 °C), the pellet was resuspended in 15% dextran in DPBS (Gibco Cat# 14190144) and centrifuged again

**Table 2 | Tissue used for proteomics analysis**

| BBN ID | PMI | Brain area | Known clinical diagnoses | Cause of death |
|---|---|---|---|---|
| BBN001.30972 | 99 | BA10 | No known abnormality | Ischemic heart disease, coronary artery atherosclerosis |
| BBN_19591 | 77 | BA9 | Asthma | Bronchial asthma |
| BBN001.29466 | 76 | BA46 | Depressive episode, unspecified; Anxiety disorder, unspecified; Other symptoms and signs involving emotional state; Personality disorder, unspecified | Suspension by ligature |
| BBN001.35823 | 72 | BA10 | Horner syndrome; Other congenital malformations of aorta | Unascertained |

Human subjects were chosen based on the following criteria: age 20–40 years, postmortem interval (PMI) < 100 h, no co-occurring neuropathology (e.g. Lewy body, TDP-43) and no co-occurring neurodegenerative disease (e.g. Parkinson's disease, multiple sclerosis). For the experiments listed in this table two males and two females between 34 and 40 years old were included.

$(10,000 \times g/15 \, min/4 \, °C)$ to yield a pellet containing blood vessels. The pellet was resuspended in DPBS, transferred to a 40 µm cell strainer, and washed with 10 mL of DPBS. From this step, the microvessels were either processed for immunofluorescence or proteomics.

*To deplete isolated vessels of endfeet,* we adapted a published protocol[101]. The entire previously frozen cortex was homogenized in 3 mL of Buffer 1 (B1; 10 mM HEPES in HBSS) after being cut into 2 mm pieces with a scalpel and centrifuged at $600 \times g/5 \, min/4 \, °C$. The resulting pellet was resuspended in 5 ml of B1 containing 18.75 µg/ml of Liberase DL (Roche, Cat # 05401054001) together with 40 U/mL of DNaseI (Invitrogen, Cat# 18047019) and incubate for 15 min at 37 °C while gently mixed every 2 min. The digested samples were moved on ice and gently homogenated using a 10 ml stripette. The homogenate was then centrifuged at $2000 \times g/5 \, min/4 \, °C$ and the supernatant resuspended in 2 mL of MCDB-131 media (Gibco Cat# 10372019) and processed for microvessel isolation as described above.

*For immunofluorescence:* The microvessels were fixed by placing the cell strainer in 10% formalin for 15 min with gentle shaking. Fixed microvessels were collected by reversing the cell strainer and applying 1% BSA in DPBS (Merckmillipore Cat# 126609) to the surface. The solution was centrifuged $(2000 \times g/10 \, min/4 \, °C)$, the pellet resuspended in DPBS, added dropwise to microscope slides and allowed to dry. Slides were stored at −80 °C until further use.

*For proteomics:* To collect microvessels, the cell strainer was reversed and 0.5% BSA in MCDB-131 was applied to the surface. Samples were centrifuged $(4100 \times g/15 \, min/4 \, °C)$ and the vessel-containing pellet resuspended in lysis buffer A (50 mM HEPES, 150 mM NaCl, 1 mM EDTA, 1X Halt protease inhibitor (ThermoFisher Cat# 78430)).

**Purification of biotinylated proteins**
Tissue (hemicortices for whole astrocyte protein purification or isolated microvessels for endfoot protein purification – see above) was homogenized using a dounce tissue grinder (Kimble Cat# 885303-002) in lysis buffer A (50 mM HEPES, 150 mM NaCl, 1 mM EDTA, 1X Halt protease inhibitor (ThermoFisher Cat# 78430)). An equivalent volume of lysis buffer B (50 mM HEPES, 150 mM NaCl, 1 mM EDTA, 0.4% SDS, 2% Triton X-100, 2% Na-deoxycholate) was added after 5 min. The samples were then sonicated (30 s on, 30 s off; for whole astrocyte – 60 cycles, for endfoot – 10 cycles at 4 °C) in a Bioruptor® Pico sonication device (Diagenode Cat# B01060010). Sonicated homogenates were centrifuged $(15,000 \times g/15 \, min/4 \, °C)$ and the supernatant ultracentrifuged $(100,000 \times g/45 \, min/4 \, °C;$ Beckman-Coulter Optima Max-XP). Cleared supernatant was supplemented with SDS to a final concentration of 1%. Samples were heated at 95 °C for 5 min. Streptavidin-coated magnetic beads (ThermoFisher Cat# 88817) were combined with the samples and incubated overnight at 4 °C with agitation. Bound beads were subsequentially washed with 2% SDS wash buffer (1% Triton X-100, 1% Na-deoxycholate, 25 mM LiCl), 1 M NaCl and 50 mM triethylammonium bicarbonate (Sigma Cat# T7408). All steps were performed on ice or at 4 °C. After the last wash, beads were resuspended in 50 mM triethylammonium bicarbonate. The samples were snap-frozen and stored at −80 °C until further use.

Samples from both Astrocyte-TurboID or Astrocyte-tdTomato injected animals underwent this process. We used Astrocyte-tdTomato as a negative control because it accounts for three potential issues – (1) binding of endogenously biotinylated proteins to the streptavidin beads, (2) potential additional endogenous protein biotinylation due to the biotin injected to the mice, (3) non-specific protein binding to the streptavidin beads.

**On-bead protein digestion**
Streptavidin beads were washed three times with ice-cold wash buffer (20 mM HEPES pH 7.5, 150 mM NaCl) and resuspended in 20 mM HEPES. Bead-bound proteins were reduced by adding 10 mM TCEP (Tris(2-carboxyethyl)phosphine hydrochloride; Sigma Cat# C4706) and incubated on a shaking thermomixer (Eppendorf; 1000 RPM/30 min/56 °C). To alkylate the proteins, 2 M urea (Thermo Scientific Cat# 140750010) and 20 mM iodoacetamide (Sigma Cat# I1149) were added and incubated on a thermomixer (1000 RPM/30 min/RT).

To digest the proteins, beads were incubated with 200 ng trypsin +Lys-C (Thermo Scientific A40007) on a thermomixer (1250 RPM/1 h/37 °C). The supernatant was transferred to a new tube and beads were resuspended in an equal volume of 50 mM triethylammonium bicarbonate. To both the supernatant and beads, an additional 200 ng trypsin+Lys-C was added and incubated on a thermomixer (1250 RPM/overnight/37 °C). Resulting supernatant and bead digests were combined before proceeding. The beads were centrifuged $(2500 \times g/2 \, min / RT)$, supernatant collected and trypsin digestion quenched by adding 1% trifluoroacetic acid (Fisher Scientific, A116) and centrifuging at max speed for 10 min.

Peptides were isolated using homemade stage tips. A single 16-gauge SDB-RPS disk (3 M Empore #2241; SDB-RPS, polystyrenedivinylbenzene-reversed phase sulfonate, 12 µm particle size, 47 mm, CDS analyticals) was isolated using a flat needle (Sigma, Cat# Z261378) and blown into a 250 µL pipette tip with compressed air. The stage tips were activated by adding wash buffer 1 (1% trifluoroacetic acid in 100% mass spectrometry-grade isopropanol) and centrifuging $(1500 \times g/5 \, min/RT)$. The peptides were twice added to the activated stage tip and centrifuged $(1500 \times g/5 \, min/RT)$ to ensure binding. The bound peptides were washed with wash buffer 1 (1% trifluoroacetic acid in 100% mass spectrometry-grade isopropanol) and wash buffer 2 (0.2% trifluoroacetic acid in 3% acetonitrile). The peptides were eluted by subsequent additions of elution buffer 1 (1.25% ammonium hydroxide in 50% acetonitrile) and elution buffer 2 (1.25% ammonium hydroxide in 80% acetonitrile). The eluted peptides were stored at −20 °C until further use. Prior to LC-MS/MS analysis, samples were evaporated to dryness (Savant SPD140DDA concentrator with a Savant RVT5105 refrigerated vapor trap). All used solutions were LC-MS/MS grade.

**Human vessel isolation and protein digestion**
As described above for mouse but with the following changes:

Unfixed, snap-frozen cortical human brain tissue (Table 2 and Supplementary Data 6) was sourced from Edinburgh Brain and Tissue Bank, a Medical Research Council funded facility with research ethics

committee (REC) under the ethical approval (21/ES/0087). Informed consent was either obtained from participants in life and/or from the nearest relative for deceased participants, in keeping with legal requirements. Sex information is based on clinical data. 250–800 mg of tissue was used for vessel isolation. Three pieces from each sample (i.e. bulk tissue samples) were punched out with a 1.5 mm biopsy punch (Kai, Cat# BPP-15F) and placed directly into lysis buffer (50 mM HEPES pH 7.5, 150 mM NaCl, 1 mM EDTA, 0.2% SDS, 1% Triton X-100, 1% sodium deoxycholate, 1X Halt protease inhibitor (ThermoFisher, Cat# 78430)). Remaining tissue was used for vessel isolation. All samples were homogenized with stainless steel beads (Biospec, Cat# 11079132ss) in a bead homogenizer (Bertin, Cat# P000669-PR240-A). To obtain the microvessel pellet, vessel homogenate was centrifuged in MCDB-131 media (2000 × $g$/5 min/4 °C) and then in 15% dextran (4200 × $g$/45 min/4 °C).

Human microvessels captured on a 40 µm cell strainer were lysed with the direct addition of lysis buffer. The resulting vessel lysate and bulk tissue homogenate were sonicated (30 s on, 30 s off for 10 cycles at 4 °C) in a Bioruptor® Pico sonication device (Diagenode, Cat# B01060010) and ultracentrifuged (100,000 × $g$/45 min/4 °C; Beckman-Coulter Optima Max-XP). Proteins were precipitated with the addition of 20% (v/v) trichloroacetic acid (Merck, Cat# 91228-100 G), washed three times with ice-cold acetone, dried and resuspended in DLT buffer (50 mM TEAB, 0.5% sodium deoxycholate, 12 mM sodium N-lauroylsarcosine). Samples were snap-frozen and stored at −80 °C until further use. Isolated protein content was quantified with BCA assay (ThermoScientific, Cat# 23225). All steps were performed on ice unless otherwise indicated.

Isolated proteins were reduced with the addition of 10 mM TCEP (Sigma Cat# C4706) and incubated on a thermomixer (1250 RPM/10 min/60 °C), then alkylated by adding 40 mM iodoacetamide (Sigma Cat# I1149) and incubated on a thermomixer (1250 RPM/40 min/RT). Finally, 5% (v/v) SDS, 1.2% (v/v) phosphoric acid and 6 volumes of S-TRAP buffer (90% (v/v) methanol in 100 mM triethylammonium bicarbonate) were added.

Proteins were bound to S-TRAP micro columns (Protifi, Cat# C02-micro), washed four times with S-TRAP buffer, topped with 0.5–1 µg trypsin/Lys-C (Thermo Scientific, A40007) and incubated overnight at 37 °C. Digested peptides were eluted with one wash with 50 mM TEAB, one wash with elution buffer I (0.2% (v/v) formic acid), and three washes with elution buffer II (0.2% (v/v) formic acid, 50% (v/v) acetonitrile). Digested peptides were snap-frozen and stored at −80 °C until further use. Prior to LC-MS/MS analysis, the solution was evaporated to dryness (Savant SPD140DDA concentrator with a Savant RVT5105 refrigerated vapor trap). All used solutions were LC-MS/MS grade.

## LC-MS/MS

For mouse samples, peptides were resuspended in 0.1% formic acid prior to EvoTip (EvoSep, Odense, Denmark) preparation. EvoTips were prepared according to the manufacturer's instructions. Briefly, Evo-Tips were washed with 0.1% formic acid in acetonitrile, conditioned with 1-propanol, and equilibrated with 0.1% formic acid. Samples were loaded onto the tip and washed 4 times with 0.1% formic acid. A preservation reservoir of 100 µL 0.1% formic acid was left in the tip to avoid drying out. 10% of the astrocyte and 50% of the endfoot peptide sample volume were injected into a Bruker timsTOF SCP mass spectrometer using an EvoSep One LC system, using the "30 samples per day" method with a 44-min gradient. The analytical column used was an "endurance" column (C18, 1.9 µm, 15 cm × 150 µm; EV1106). Data was acquired in the standard DIA PASEF template mode.

For human samples, peptides were analyzed on a Orbitrap Fusion Lumos Tribrid mass spectrometer interfaced with a 3000 RSLC nano liquid chromatography system. 1 µg of each sample was loaded on to a 2 cm trapping column (PepMap C18 100 A – 300 mm, Thermo Fisher Scientific Cat# 160454) at a 5 mL/min flow rate using a loading pump. Samples were analyzed on a 50 cm analytical column (EASY-Spray

column, 50 cm 75 mm ID, Cat# ES803) at a 300 nL/min flow rate that is interfaced to the mass spectrometer using Easy nLC source and electrosprayed directly into the mass spectrometer. A linear gradient between solvent A (0.1% formic acid in water) and solvent B (0.1% formic acid in acetonitrile) was established as such: 3% to 30% of solvent B at a 300 nL/min flow rate for 105 min; 30% to 40% solvent B for 15 min; 35–99% solvent B for 5 min, which was maintained at 90% B for 10 min. The column was washed with 3% solvent B for another 10 min comprising a total 140 min run with a 120 min gradient in a data independent acquisition (DIA) mode. 45 isolated 350–1500 m/z windows were used. Collision induced dissociation (CID) was used for fragmentation of peptides.

## Proteomics data analyses

Raw mass spectra files were analyzed using DIA-NN v.1.8.1[102] in library free mode. For mouse samples, peptides were searched against a Uniprot mouse database (downloaded March 2022) supplemented with TurboID and tdTomato sequences. For human samples, peptides were searched against a Uniprot reference human proteome (downloaded March 2023). Both mouse and human peptides were also searched against a custom database of common contaminants based on the cRAP database (https://www.thegpm.org/crap/). The default parameters for double pass neural network classifier were used with the following exceptions: the mass accuracy and MS1 accuracy was set to 10.0, and "Heuristic protein inference" was unselected. Precursor false discovery rate was set to 0.1.

All proteomics data presented here originates from the "report.pg_matrix.tsv" output file from DIA-NN. All analysis was conducted using R (R Core Team, 2023), RStudio (RStudio Team, 2023), and the tidyverse package[103] (version 2.0.0).

Ambiguous, species-mismatched, contaminant proteins, and proteins not mapped to genes were removed from analysis. For comparisons between mice expressing TurboID and tdTomato AAV, proteins found in at least 33% of the samples from either astrocyte or endfoot protein isolations regardless of AAV were kept for further analysis due to the inherent difference in isolated proteins between TurboID and tdTomato samples. For comparisons between LPS and PBS samples, which only received TurboID AAV and are therefore expected to yield a similar number of isolated proteins between conditions, proteins were kept for further analysis if they were found in at least 50% of both PBS and LPS samples. Human proteins were kept if they appeared in at least 2 samples from bulk or at least 2 samples from vessel protein isolations. Quantile normalization was performed using the *preprocessCore* package[104] (version 1.66.0). Mouse astrocyte and endfoot datasets were normalized separately, human bulk and vessel datasets were normalized together. For mouse samples, differential expression analysis was conducted with the package *limma*[105] (version 3.56.2). If a protein was not observed in a sample, its abundance was imputed with a value of 1. Differential expression was considered significant when $p$-value < 0.05. For human samples, differential expression analysis was conducted with the package *DEP*[106] (version 1.26.0), no imputation was used, and differential expression was considered significant when $p$-value < 0.05. Gene ontology and pathway analyses were generated using Qiagen IPA[58].

For Supplementary Fig. 2b, c, all peptides mapped to dystrophin (DMD) were extracted by filtering the DIANN output "report.tsv" for peptides found in the PBS endfoot samples. A total of 30 peptides were identified and the number of times each was recorded. The peptide sequences were manually mapped to the FASTA sequences for DMD and isoforms Dp71 and Dp40, which were obtained from NCBI. The FASTA sequence for mouse Dp140 is not annotated but is ~1200 amino acids (AA) long in humans and conserved through the C-terminus, so the same was assumed for this analysis. The AA positions of each peptide sequence was mapped onto the full-length DMD sequence and counted to generate the graphs.

For Supplementary Fig. 3, we compared our endfoot proteome to previously published datasets[36–39]. For each published dataset, we used the provided data with some additional filters. For Stokum et al., we filtered the data for endfoot enriched proteins (log(endfoot/cell body) > 0). For Kameyama et al., we used the proteins unique to the purified perivascular astrocyte endfeet (PV-AEF) and the proteins enriched in purified PV-AEF identified by the following filters: a) enriched in the crude PV-AEF fraction (CA/B0 > 0), b) enriched in the pure PV-AEF fraction (PA/B0 > 0), c) ANOVA $q$-value < 0.05, d) removed rows with multiple gene names separated by a semicolon. For Alonso-Gardon et al., no additional filtering was applied; we used *Hepacam* (HECAM), *Mlc1* (MLC1), and all identified proteins. For Soto et al., we used the proteins found to be enriched and unique to AQP4-BioID2 versus AQP4-GFP (1) or astrocyte BioID2 (2) and removed proteins with no gene name. Plots were generated using the UpSetR package[107]. To directly compare the endfoot and astrocyte proteomes (PBS and TurboID AAV), a relative normalized abundance for each protein shared between the endfoot and astrocyte proteomes (1151 proteins) was generated using the "report.stats" output from DIANN. For each sample, the TotalQuantity value (reflecting the overall protein abundance per sample) was extracted, and divided by the highest value. This sample-specific normalization factor was used to scale the abundance of each shared protein (1151) protein in each sample. The resulting normalized values were used for differential expression analysis with Limma, as described above.

For Supplementary Fig. 4, transporters were identified by comparing the endfoot proteome to a database of SLCs (https://slc.bioparadigms.org/).

For Fig. 2i, receptors in endfeet were identified by comparing receptors from CellTalkDB[61] (http://tcm.zju.edu.cn/celltalkdb/) with the endfoot proteome and manually curating the list to select only the ones with known downstream molecular cascades.

For Supplementary Figs. 2a, 3b, 6c, and 11d, cell type markers were included due to their widespread acceptance as cell type markers in immunohistochemical and/or western blot applications. Additional markers were identified in single-cell RNA-seq datasets of bulk brain tissue (UCSC Cell Browser v1.2.12[108]) and isolated vasculature[73], and review manuscripts[109,110].

### Quantitative real time-polymerase chain reaction (qRT-PCR)

To check cell specificity of the Cdh5-CRE/RiboTag mouse line, purified mRNA from Input and IP samples was transcribed to cDNA using Superscript IV reverse transcriptase (ThermoFisher Cat# 18090010). Cell markers abundance was assessed by qRT-PCR using Fast SYBR Green (ThermoFisher Cat# 4385612) and previously published primers[111]. Gene expression was calculated relative to the housekeeping *RplpO* gene based on their Ct values using the formula: $2^{-\Delta Ct}$ ($\Delta Ct = Ct$ (gene of interest) − Ct (housekeeping gene)).

### Mouse whole tissue and BEC-specific RNA-seq and differential expression analysis

Mice were culled by cervical dislocation and the brain was quickly removed and cut into 1.5 mm coronal slices using a rodent brain matrix. To obtain enough RNA, each biological sample contained PFCs from two mice. The detailed protocol has been published[111]. In brief, brain slices were placed in cold RNAse-free PBS and PFC was further dissected and collected for RNA extraction. The dissected tissue was homogenized using a dounce tissue grinder (Kimble Cat# 885303-002) in 1 mL of homogenization buffer (47 mM Tris, 94 mM KCl, 11.3 mM MgCl$_2$, 1%NP-40, 1 mM DTT, protease inhibitor (Sigma Cat# P8340), 200 U/mL of RNasin (Promega Cat# N2115), 100 µg/mL cycloheximide, 1 mg/mL heparin). The homogenate was centrifuged (10,000 × $g$/10′/4 °C) to obtain a clear lysate, 10% (100 µL) of which was used to extract whole tissue RNA (Input). The remaining lysate (IP) was incubated with mouse anti HA-antibody (1:200, Biolegend Cat#

901514) for 4 h at 4 °C in rotation, followed by addition of 200 µL of Pierce protein A/G magnetic beads (ThermoFisher Cat#88803) and incubation overnight at 4 °C in rotation. The next day, beads were washed three times using high salt solution (50 mM Tris, 300 mM KCl, 12 mM MgCl$_2$, 1% NP-40, 1 mM DTT and 100 µg/mL). RNA was purified using RNeasy Plus Micro kit (Qiagen Cat# 74004). The concentration and quality of the RNA was assessed with an Agilent 2100 Bioanalyzer and only RNA samples with a concentration higher than 4 µg/µL and an RNA integrity number (RIN) greater than 7 were further sequenced. Sequencing libraries were prepared using TruSeq RNA stranded mRNA kit (Illumina Cat# 20020594) and libraries were sequenced on a NextSeq 500 system (Illumina).

Reads were mapped to the primary assembly of the mouse reference genome contained in Ensembl release 104, using the STAR RNA-seq aligner, version 2.7.9a[112]. Tables of per-gene read counts were then generated from the mapped reads with featureCounts, version 2.0.2[113]. Differential gene expression was performed in R using DESeq2, version 1.30.1[114].

### Identification of BEC-endfoot ligand-receptor pairs

The full list of mouse ligand-receptor pairs was downloaded from CellTalkDB[61] (http://tcm.zju.edu.cn/celltalkdb/). Ligands were identified by comparing the genes identified in BEC RNA-seq data to CellTalkDB. For ligands at baseline (PBS), CellTalkDB ligands were compared to genes identified with FPKM > 1 for BEC IP in PBS samples. For ligands altered by LPS, CellTalkDB ligands were compared to upregulated (LPSvsPBS, FPKM > 1, logFoldChange >1, FDR < 0.05) or downregulated genes (LPSvsPBS, FPKM > 1, logFoldChange < −1, FDR < 0.05). Receptors in endfeet were identified as described above. Ligand-receptor pairs were identified by comparing ligands found in BEC RNA-seq with receptors found in the endfoot proteome. For PBS ligand-receptor pairs, only ligands and receptors identified in the PBS samples were considered. For ligand-receptor pairs that change with LPS, up- or downregulated ligands after LPS were matched with receptors found in the endfoot proteome samples receiving either PBS or LPS.

### Comparison of ligand-receptor pairs involved in mouse BEC-endfoot communication against human data

To evaluate the overlap of signaling pathways identified between mouse BECs and endfeet with human data, we generated our own young adult (34–40-years-old) healthy human vessel dataset (Fig. 7, Supplementary Fig. 11 and Supplementary Data 6), and sourced three previously published datasets on neurodegeneration-associated molecular changes in the human vasculature: single-nucleus RNA sequencing (snRNA-seq) of multiple sclerosis (MS)[71], snRNA-seq of Alzheimer's disease (AD) patients[73] and proteomics of human vasculature isolated from AD patients and controls[72] (Supplementary Data 7). To analyze single-cell data from Yang et al. and Macnair et al., per-cell, per-gene count matrices were loaded using Seurat[115] (R package version 4.3.0). Pseudo-bulk differential expression analysis was then performed by summarizing single cell gene expression profiles for different cell types at the sample level using the aggregate-BioVar (https://github.com/jasonratcliff/aggregateBioVar; R package version 1.6.0), then differentially expressed genes between conditions were calculated using DESeq2[114] (R package version 1.36.0). To analyse proteomics data from Wojtas et al., we used publicly available data. In downstream analysis, we focused on differential expression between cortical gray matter demyelinated lesions and controls in the Macnair et al. study, and between cortex of Alzheimer's disease patients and controls in the Yang et al. and Wojtas et al. studies. A ligand-receptor pair was defined as "identified" in a human dataset if both the ligand and the receptor were detected. For snRNA-seq datasets, ligands were searched in the capillary cluster and receptors in the astrocyte cluster. For proteomics datasets, ligands and receptors were searched in the

entire vascular proteome. A ligand-receptor pair was defined as "differentially expressed" in a human dataset if either the ligand or the receptor was differentially expressed (FDR < 0.1 for snRNA-seq studies, Bonferroni-adjusted $p < 0.1$ for proteomic studies) in the dataset, and its corresponding ligand or receptor was identified in the dataset. See Supplementary Data 7 for further details on the downstream analysis.

## Statistical analysis

Data were analyzed with GraphPad Prism or R. Unpaired Student's two-tailed t-test was used for Fig. 1i (parametric data). Datasets containing more than one datapoint per animal were analyzed using linear mixed-effects modeling (LMM), which included random factors to control for pseudo-replications. After assembling an initial LMM, the normality of the residuals was assessed using Shapiro-Wilk test. To meet model assumptions, data with non-normally distributed residuals were transformed using Tukey Ladder of Powers. To identify significance main effects, Type 3 ANOVA with Satterthwaite approximation was run on the LMM modifying accordingly the code available at https://github.com/Diaz-Castro-Lab/Regional-blood-brain-barrier-ageing-LMM-script. When relevant, the emmeans R package (version 1.10.7) was used for multiple comparisons by Tukey post hoc tests. For distribution analysis, a chi-squared test was used to determine significant differences between groups. No sex-based comparisons were made as this study was not powered for it. Investigators were blinded during analyses.

Representative images are based on at least three independent experiments.

Graphs were generated using both GraphPad Prism and R. Most graphs generated in R were made with the ggplot2 package (version 3.5.1). Sankey plots in Figs. 4, 7, and Supplementary Fig. 12 were made using the networkD3 (version 0.4) and htmlwidgets (version 1.6.4) packages. UpSet plots in Figs. 1, 8, and Supplementary Fig. 3 were made using the UpSetR package (version 1.4.0). Circle dendrogram in Supplementary Fig. 4 was generated the ggraph (version 2.2.1) and igraph (version 2.1.2) packages. Data are represented as mean ± standard error of the mean (SEM) or median depending on the distribution of the data and as indicated in the figure legends. Significance level was considered at $p$-value < 0.05, FDR < 0.05, FDR < 0.1 or $p_{Bonf} < 0.1$, depending on the dataset as indicated in the figure legends.

## Reporting summary

Further information on research design is available in the Nature Portfolio Reporting Summary linked to this article.

## Data availability

The mass spectrometry proteomics data have been deposited to the ProteomeXchange Consortium via the PRIDE[116] partner repository with the dataset identifier PXD056232 for the mouse dataset, and identifier PXD055837 for the human dataset. RNA-seq data was uploaded onto ArrayExpress with accession ID E-MTAB-14490. Processed proteomics and RNA-seq data can be found in the Supplementary Data. Other datasets are available upon request. Source data are provided with this paper.

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

## Acknowledgements

This work was supported by UK Dementia Research Institute (UK DRI-4007 and UK DRI-Edin006), through UK DRI Ltd, principally funded by the Medical Research Council, and The Royal Society funds (RGS \R1\231330) to B.D.C. S.H. was additionally supported by an Alzheimer's Society Postdoctoral Fellowship (Ref. 609). A.C. is a student on the Translational Neuroscience PhD Programme and is funded by Wellcome Trust (Ref: 218493/Z/19/Z). P.B.-L. was additionally supported by Alzheimer's Research UK (ARUK-ECRBF2023B-005) and Alzheimer's Society Postdoctoral Fellowship (Ref. 663). The DRA lab is supported by the UK Medical Research Council [grant number MC_UU_00018/1]. The YL lab is supported by My Name'5 Doddie foundation. We thank Viviana Gradinaru, Alice Ting, and Baljit Khakh for sharing plasmids. We thank the human brain tissue donors and their families, and the Edinburgh Brain and Tissue Bank for providing this tissue. For the purpose of open access, the author has applied a Creative Commons Attribution (CC BY) license to any Author Accepted Manuscript version arising from this submission.

## Author contributions

S.H. performed the proteomic experiments, immunofluorescence staining, data analyses, and made most of the figures. I.B.-F. performed the BEC RNA-seq experiments and data analyses. A.C. performed the human sample proteomics, immunofluorescence staining, and human ligand-receptor analyses. N.P.R. performed isolated vessel and mouse tissue immunofluorescence staining and analyses. I.R. searched for and classified the solute carrier protein family and performed immunofluorescence staining and analyses. C.C. performed RNAscope and tissue expansion experiments and quantifications. P.B.-L. performed human brain tissue immunostaining and quantifications. C.P.P. performed cell expression specificity experiments for BEC Ribotag mice and vessel isolation with endfoot depletion. I.B.-F., A.C., N.P.R., I.R., C.C., P.B.-L., and C.P.P also contributed to making figures. K.E. performed the RNA-seq differential expression analyses under the supervision of O.D. B.G. managed the LC-MS/MS runs and trained S.H. on the bioinformatics for protein identification and differential expression. R.S.N. and D.R.A. advised and supported with proteomics expertise that was key to the success of this project. D.L. and Y.L. produced the AAVs used in this manuscript. B.D.C. conceived the study, directed the study, and coordinated the collaborative work. B.D.C. and S.H. wrote the paper with substantial contribution from I.B.-F. All the authors commented on the manuscript, which was finalized by B.D.C. and S.H.

## Competing interests

The authors declare no competing interests.
