## [Transparent Peer Review file · Nature Communications]

Molecular profiling of brain endothelial cell to astrocyte endfoot communication in mouse and human

Corresponding Author: Dr Blanca Díaz Castro

Version 0:

Reviewer comments:

Reviewer #4

(Remarks to the Author)

All major and minor concerns have been adequately addressed by the authors.

There are just a few minor notes to address for clarity:

1. On lines 71-73, the authors write "In capillaries, ~85% of the BEC's abluminal side is directly facing endfeet, with only the basement membrane standing between the two4." The cited reference seems to quantify that ~63% of BECs are not covered by pericytes so only this percentage of BECs would have only the basement membrane in between the endfeet and the BEC abluminal side.
2. There is a missing parenthesis on line 413 that makes the sentence confusing.
3. On lines 617-618, the authors write that "Note to table 1: Human subjects were chosen based on the following criteria: age 20-40 years, postmortem interval (PMI) < 100 hours, no known neurological conditions." However, subject BBN001.29466 had diagnoses of a depressive episode and anxiety disorder, which could indicate functional brain differences. The criteria that the authors used for no known neurological conditions would be helpful to clarify.

Reviewer #5

(Remarks to the Author)

In this article, Hill et al. attempt to characterize ligand-receptor interactions between astrocytes and brain endothelial cells. The study addresses a long-standing gap in the literature. It is well established that astrocytes interacting with blood vessels regulate cerebral vascular functions, likely through bidirectional communication. It is also known that these interactions are altered in various neuropathologies. However, the molecular pathways linking these two compartments remain poorly understood.

The use of BioID technology in astrocytes, combined with the mechanical isolation of blood vessels to enrich for astrocyte endfeet, and the comparison with the endothelial transcriptome, represents a well-conceived and appropriate strategy. The data presented in this paper undeniably contribute to advancing the field. A notable weakness is the absence of important controls and the choice of the FZD7/Wnt10 pathway for validation. The detection experiments for both proteins are difficult to interpret. This choice does not convincingly demonstrate the value of the study.

Terminology and Clarity Issues:

- Line 72: "BEC's abluminal side is directly facing endfeet, with only the basement membrane standing between the two."

This statement is inaccurate. In many regions, astrocytes and BECs are separated by pericytes, vascular smooth muscle cells (VSMCs), fibroblasts, or perivascular macrophages.

- Reference 23: This kinase appears to have been largely ignored in the literature since its original publication. Its expression in astrocytes has not been clearly documented. Please verify this in the literature.

- Reference 25: Hmgb1 is a histone and probably does not represent a direct signaling pathway. If the goal is to cite all astrocyte-derived molecules that influence vascular function, then connexins, aquaporins, Kir channels, etc., should also be included.

Experimental Design and Definitions:

- Astrocyte endfeet are likely diverse and heterogeneous. The manuscript states that the purified MV pellet was transferred through a 40 μ m cell strainer, implying that the vascular fraction studied is not purely capillary. It is essential to clearly define the sample composition, not only using Aqp4/lectin immunofluorescence but also by quantifying the presence of other vascular cell types—ECs, VSMCs, pericytes, fibroblasts, and perivascular macrophages—perhaps through transcriptomic or qPCR profiling.
- The number of transfected astrocytes per viral condition must be reported.
- Figure 2D: How do the authors explain the high number of proteins immunoprecipitated in the tdTomato control?
- Line 128: Are the other transcriptomic databases derived from cortical tissue? Given brain regional heterogeneity, the tissue origin should be specified for each dataset used.
- DMD: As a very large protein with multiple isoforms, only a few are likely expressed in endfeet. The authors should specify which isoforms are likely present.
- Line 175: To support endothelial specificity, please report levels of VSMC-, fibroblast-, and macrophage-specific proteins in Extended Figure 5b.
- Line 186: A critical control is missing. The validity of the endfeet proteome relies on the assumption that astrocyte endfeet remain attached to blood vessels post-LPS treatment. This must be demonstrated. If LPS causes detachment, the results could reflect endfeet loss rather than molecular changes.
- Line 220: A reference is missing: S. Guérit et al., *Progress in Neurobiology*, 199, 101937 (2021), which shows that astrocyte-specific deletion of *Evi*, a Wnt secretion mediator, affects BBB maintenance.

Concerns with Validation of the FZD7/Wnt10 Pathway:

- “First, we validated the expression of FZD7 in endfeet through RNA in situ hybridization (RNAscopeTM) (Extended Data Fig. 7a).”

This statement is misleading. The experiment is supposed to demonstrate mRNA presence in endfeet, which might be unrelated with the presence of Fzd7 protein in endfeet. More critically, the images are unconvincing—most FISH signals appear in non-astrocyte cell somata. The experiment should be repeated with DAPI staining to show all cell bodies, and ideally with dtTomato-labeled astrocytes to demonstrate astrocytic expression of FZD7. Datasets from Betzolt lab (ARN) and Rurak et al 2022 (Astrocyte TRAP) suggest Fzd7 expression is extremely low in astrocytes or expressed in very few cells.

- Extended Data Fig. 7g: The FZD7 immunofluorescence appears to be background staining. The authors trace the vessel on only one side. In the PBS control, the signal is asymmetric, and the diffuse lower side signal does not colocalize with β -dystroglycan. This suggests non-specific staining.
- Figure 5A: Similar concerns—labeling is very diffuse and lacks clarity.
- To convincingly show FZD7 presence in endfeet, the authors should perform comparative IF and/or western blotting on isolated MVs with and without endfeet.
- Wnt10b is a secreted protein, likely difficult to detect. Figure 5E is also unconvincing. What does “inside the vessel” mean in this context? Is it not in pericytes?

Version 1:

Reviewer comments:

Reviewer #5

(Remarks to the Author)

Thank you very much for responding so well to all my comments.
This article is excellent.

We are very thankful for the time and attention the reviewers have dedicated to assess our manuscript. Their suggestions have reinforced our conclusions and we now present a stronger manuscript.

REVIEWER COMMENTS

Reviewer #4 (Remarks to the Author):

All major and minor concerns have been adequately addressed by the authors. There are just a few minor notes to address for clarity:

1. On lines 71-73, the authors write “In capillaries, ~85% of the BEC’s abluminal side is directly facing endfeet, with only the basement membrane standing between the two.” The cited reference seems to quantify that ~63% of BECs are not covered by pericytes so only this percentage of BECs would have only the basement membrane in between the endfeet and the BEC abluminal side.

This is right! We apologize for the oversight. We have corrected it now to 63% and added more references. In lines 70-72 we wrote “In capillaries, the abluminal side of the BECs faces both endfeet and pericytes, with most of its perimeter (~63%) adjacent to endfeet, separated from them only by the basement membrane”.

2. There is a missing parenthesis on line 413 that makes the sentence confusing.

Thank you for pointing this out. We have corrected it now (line 435).

3. On lines 617-618, the authors write that “Note to table 1: Human subjects were chosen based on the following criteria: age 20-40 years, postmortem interval (PMI) < 100 hours, no known neurological conditions.” However, subject BBN001.29466 had diagnoses of a depressive episode and anxiety disorder, which could indicate functional brain differences. The criteria that the authors used for no known neurological conditions would be helpful to clarify.

We apologize for the lack of clarity. We have changed the Note to tables 1 and 2 to “Human subjects were chosen based on the following criteria: age 20-40 years, postmortem interval (PMI) < 100 hours, no co-occurring neuropathology (e.g. Lewy body, TDP-43) and no co-occurring neurodegenerative disease (e.g. Parkinson’s disease, multiple sclerosis).” (lines 648-650 and 926-928).

Reviewer #5 (Remarks to the Author):

In this article, Hill et al. attempt to characterize ligand-receptor interactions between astrocytes and brain endothelial cells. The study addresses a long-standing gap in the literature. It is well established that astrocytes interacting with blood vessels regulate cerebral vascular functions, likely through bidirectional communication. It is also known that these interactions are altered in various neuropathologies. However, the molecular pathways linking these two compartments remain poorly understood.

The use of BioID technology in astrocytes, combined with the mechanical isolation of blood vessels to enrich for astrocyte endfeet, and the comparison with the endothelial transcriptome, represents a well-conceived and appropriate strategy. The data presented in this paper undeniably contribute to advancing the field. A notable weakness is the absence of important controls and the choice of the FZD7/Wnt10 pathway for validation. The detection experiments for both proteins are difficult to interpret. This choice does not convincingly demonstrate the value of the study.

Terminology and Clarity Issues:

- Line 72: "BEC's abluminal side is directly facing endfeet, with only the basement membrane standing between the two."

This statement is inaccurate. In many regions, astrocytes and BECs are separated by pericytes, vascular smooth muscle cells (VSMCs), fibroblasts, or perivascular macrophages.

We apologize for the lack of clarity here. We have rephrased the sentence to take into consideration pericytes too. We have focused only on capillaries in this introduction because it is where we think the direct communication between endfeet and BECs can occur. As the reviewer states, we would not expect much endfoot-BEC communication in bigger caliber vessels due to the position of VSMCs, fibroblasts and perivascular macrophages between them. In capillaries, however, there are parts in which the pericytes are between endfeet and BECs, and parts where they are "directly facing each other, with only the basement membrane between them"; it is in these regions where we think they could communicate.

In lines 70-72, this sentence now reads "In capillaries, the abluminal side of the BECs faces both endfeet and pericytes, with most of its perimeter (~63%) adjacent to endfeet, separated from them only by the basement membrane".

- Reference 23: This kinase appears to have been largely ignored in the literature since its original publication. Its expression in astrocytes has not been clearly documented. Please verify this in the literature.

We have removed this citation.

- Reference 25: Hmgb1 is a histone and probably does not represent a direct signaling pathway. If the goal is to cite all astrocyte-derived molecules that influence vascular function, then connexins, aquaporins, Kir channels, etc., should also be included.

HMGB1 was initially described as a chromatin-associated non-histone protein. However, throughout the years it has been recognized that HMGB1 has many roles outside the nucleus, including in the extracellular space. It is particularly well known as a secreted molecule involved in immune responses. Please see (Tang *et al*, 2023) for details.

Our intention is just to mention astrocyte ligands that could be received by BECs – "signaling molecules" rather than any protein that could indirectly modulate BEC functions. Thus, we

have kept the HMGB1 reference in the introduction (line 74; (Hayakawa *et al*, 2012)), have removed (Mishra *et al*, 2016) because it is focused on astrocyte-pericyte and not astrocyte-BEC communication, and have refrained from adding more references.

Experimental Design and Definitions:

- Astrocyte endfeet are likely diverse and heterogeneous. The manuscript states that the purified MV pellet was transferred through a 40 μm cell strainer, implying that the vascular fraction studied is not purely capillary. It is essential to clearly define the sample composition, not only using Aqp4/lectin immunofluorescence but also by quantifying the presence of other vascular cell types—ECs, VSMCs, pericytes, fibroblasts, and perivascular macrophages—perhaps through transcriptomic or qPCR profiling.

We appreciate the reviewer's comment and understand that they are requesting clarification on the type of endfeet included in our preparation – relative to the vessel types they associate with – and that contribute to the composition of our endfoot proteome. We agree with the reviewer that this is an important point, as it is likely that the endfoot functions and molecular profiles are diverse depending on the type of vessel they surround. While we recognize the utility of qPCR in assessing gene expression, we note that it may not reliably quantify the relative abundance of different vessel types, as marker expression levels do not directly correspond to cell or vessel proportions, but rather to the RNA abundance of that specific marker.

We concluded immunohistochemistry was the most accurate way to identify the different vascular segments in our purified vessel samples: arteries, arterioles, capillaries, venules and veins. We first classified the purified vessels based on size: $<10\ \mu\text{m}$ for capillaries, $10\text{--}30\ \mu\text{m}$ for arterioles and venules, and $>30\ \mu\text{m}$ for arteries and veins (Fig. 1j and 1l). This indicated that 69% of the area occupied by vessels in a given preparation (Fig. 1h) belonged to capillaries. Next, we calculated the endfoot coverage of purified vessels using AQP4 staining and classifying the vessel types based on their diameter and αSMA and lectin staining patterns. We discovered that the coverage of endfeet was higher in capillaries than in any other vessel type (Fig. 1k-m). This is expected given that, in vessels of higher caliber, the Virchow–Robin space separates the astrocyte endfeet from the other vascular cells. Finally, we calculated a score for the endfoot composition of our isolated vessels by multiplying the % of vessels of certain diameter with the mean of their endfoot coverage. This analysis suggests that from all the endfeet contributing to the endfoot proteome, the vast majority (83%) are coming from capillaries. We described these results in lines 117-123 and include the new figure below.

- The number of transfected astrocytes per viral condition must be reported.

Thank you for the suggestion, we have added this in the figure legend of Supplementary Fig. 1a (“From 360 S100 β cells counted, 152 were streptavidin positive”) and Supplementary Fig. 1d (“From 345 S100 β cells counted, 134 were tdTomato positive”).

- Figure 2D: How do the authors explain the high number of proteins immunoprecipitated in the tdTomato control?

A high number of proteins in the tdTomato control is expected based on previous studies. For example, in (Soto *et al.*, 2023) the authors also identified thousands of proteins in the GFP negative controls – shown here with two Venn diagrams we generated using their own datasets:

The high number of proteins in the negative controls is due to non-specific binding of proteins to the streptavidin-coated magnetic beads used to pull down biotinylated proteins. For this reason, the inclusion of negative controls is essential when using proximity biotinylation assays to define the proteome of a cell, or subcellular compartment. Only by focusing on proteins that are enriched over this negative control we can have high certainty that we are capturing cell specific biotinylated proteins (Fig. 2d-g). In agreement with a higher affinity of biotinylated proteins for the streptavidin beads than non-biotinylated proteins, the majority of proteins we identified are enriched in the TurboID samples over TdTomato:

To address similar concerns from other readers we have added two new sentences in the discussion (lines 448-451) – “Non-biotinylated proteins can bind non-specifically to the streptavidin-coated magnetic beads (Fig. 2c). Therefore, by concentrating on proteins that are specifically enriched in TurboID samples relative to tdTomato controls, we can confidently identify proteins that are specifically biotinylated in astrocytes or endfeet.”.

- Line 128: Are the other transcriptomic databases derived from cortical tissue? Given brain regional heterogeneity, the tissue origin should be specified for each dataset used.

We have added this information in the figure legend of Fig. 1b, Supplementary Fig. 3 and in the Supplementary Table 3 readme tab. In summary, all the published datasets were generated using whole brain, except for Soto et. al. that used striatum. It is unlikely that the protein differences we see between Soto et. al. and ours are due to the use of different brain regions, because the endfoot proteins non-detected in their experiment have been shown to be expressed in striatal endfeet or astrocytes in previous work (Diaz-Castro *et al*, 2019; Gokce *et al*, 2016; Ollivier *et al*, 2024).

- DMD: As a very large protein with multiple isoforms, only a few are likely expressed in endfeet. The authors should specify which isoforms are likely present.

Thank you for the suggestion. It has been shown that DMD has several isoforms in the brain which expression is controlled by different promoters (Tetorou *et al*, 2025). It is well described that the isoform enriched in endfeet is Dp71 (Haenggi *et al*, 2004; Nicchia *et al*, 2008). Using peptide data from our endfoot proteomics analysis, we identified the regions within the full DMD gene sequence where the detected peptides align. With this information, we were able to observe a clear enrichment of peptides that localize to the section of the gene corresponding to Dp71, further demonstrating we have an enrichment of endfoot proteins in our preparation (Supplementary Fig. 2b-c). A few peptides aligned with the Dp427 isoform, which has been shown to be expressed in arterioles (Belmaati Cherkaoui *et al*, 2021), so this is concordant with the presence of a fraction of arteriolar endfeet in our protein purification as shown above and in Fig. 1k-n. We have included this information in the results section (lines 134-143).

• Line 175: To support endothelial specificity, please report levels of VSMC-, fibroblast-, and macrophage-specific proteins in Supplementary Figure 5b.

We apologize for not showing this before. This has been added to the now Supplementary Figure 6c (see below).

• Line 186: A critical control is missing. The validity of the endfeet proteome relies on the assumption that astrocyte endfeet remain attached to blood vessels post-LPS treatment. This must be demonstrated. If LPS causes detachment, the results could reflect endfeet loss rather than molecular changes.

We thank the reviewer for the suggestion. We have calculated the endfoot coverage per vessel type, based on AQP4 signal, in both PBS and LPS injected mice and found no differences between the two conditions (Supplementary Fig. 7d-j). This has been mentioned in line 205 and we include the figure below for the reviewer's perusal.

- Line 220: A reference is missing: S. Guérit et al., *Progress in Neurobiology*, 199, 101937 (2021), which shows that astrocyte-specific deletion of Evi, a Wnt secretion mediator, affects BBB maintenance.

Thank you for the suggestion. We have added it now mentioning the role of WNT in BBB maintenance as well (line 239).

Concerns with Validation of the FZD7/Wnt10 Pathway:

- “First, we validated the expression of FZD7 in endfeet through RNA in situ hybridization (RNAscope™) (Extended Data Fig. 7a).”

This statement is misleading. The experiment is supposed to demonstrate mRNA presence in endfeet, which might be unrelated with the presence of Fzd7 protein in endfeet. More critically, the images are unconvincing—most FISH signals appear in non-astrocyte cell somata. The experiment should be repeated with DAPI staining to show all cell bodies, and ideally with dtTomato-labeled astrocytes to demonstrate astrocytic expression of FZD7. Datasets from Betzolt lab (ARN) and Rurak et al 2022 (Astrocyte TRAP) suggest Fzd7 expression is extremely low in astrocytes or expressed in very few cells.

We fully agree with the reviewer that RNAscope is not the best way to demonstrate the expression of FZD7 in astrocytes. As mentioned, RNAseq datasets show very low RNA levels of FZD7 in astrocytes. In addition, RNAscope is tricky for this specific question because, although endfeet present local translation (Boulay *et al*, 2017), the field does not know yet if all endfoot proteins are locally translated or some of them are produced in the cell body of the astrocytes and then transported to the endfoot (line 378). All of this, in addition to the fact that absolute RNA level does not always correlate with protein level (Chai *et al*, 2017; Ghazalpour *et al*, 2011; Wang *et al*, 2019)(line 379), made us reconsider the use of RNAscope to demonstrate the expression of FZD7 in astrocyte endfeet. Thus, we have made additional efforts to demonstrate the presence of FZD7 in endfeet at the protein level (see below) and we have removed the FZD7 RNAscope data from the manuscript, as it is not conclusive.

- Extended Data Fig. 7g: The FZD7 immunofluorescence appears to be background staining. The authors trace the vessel on only one side. In the PBS control, the signal is asymmetric, and the diffuse lower side signal does not colocalize with β -dystroglycan. This suggests non-specific staining.
- Figure 5A: Similar concerns—labeling is very diffuse and lacks clarity.

To address the two points raised above by the reviewer, we have performed tissue expansion experiments together with confocal super-resolution microscopy (Zeiss Airyscan). Expanded WT mouse cortical tissue expressing GFP in astrocytes, was stained for FZD7, GFP and vessels with lectin. We quantified the signal overlap of FZD7 with GFP positive endfeet or lectin in cross-sections of capillaries, and we could clearly observe the presence of FZD7 in endfeet and its depletion in the vasculature. This data is displayed in Figure 5a-b (lines 246-251) and we include it below.

In addition, we include in this response additional images from 3 more mice for the reviewer to be able to examine more examples.

Moreover, we have substituted the images in Fig. 6a to show the expression of FZD7 in endfeet in not expanded tissue - used for FZD7 quantification - with more clarity (below).

• To convincingly show FZD7 presence in endfeet, the authors should perform comparative IF and/or western blotting on isolated MVs with and without endfeet.

We are very grateful for this suggestion; we think it has greatly contributed to convincingly show the localization of FZD7 in endfeet. As suggested, we isolated vessels with or without depletion of endfeet and performed immunohistochemistry of FZD7, AQP4 and lectin (Fig. 5c-e). We performed two types of quantifications, one to calculate the distance between the FZD7 signal with AQP4 or lectin (Fig. 5f-g), and another one to calculate the loss of signal of each marker upon endfoot depletion (Fig. 5h-i). We have described the results in lines 251-256) and we attach the figure below.

- *Wnt10b* is a secreted protein, likely difficult to detect. Figure 5E is also unconvincing. What does “inside the vessel” mean in this context? Is it not in pericytes?

We apologize for the lack of clarity. As pointed out, a secreted protein is difficult to detect and assign to specific cells. In terms of the cell specificity, both RNAseq and RNAscope data demonstrate *Wnt10b* RNA is localized in BECs. To further confirm this, we have repeated the RNAscope staining (in 4 mice per condition), now including the labelling of the BEC specific nuclear marker ERG, together with the *Wnt10b* probe and lectin (Supplementary Fig. 8a and below). These experiments clearly demonstrated an association of *Wnt10b* RNA with BEC nuclei.

At the protein level, due to the extracellular location of this protein, we decided to measure its expression “inside the vessel”. To do this, we measured WNT10B signal within a region of interest that contained both the vessel and the endfeet. We achieved this using β -dystroglycan staining as the endfoot boundary and measuring the WNT10B signal within those boundaries, i.e. endfoot plus vessel area. We have explained this with more clarity in the methods (line 1079) and in the figure legend (Fig. 6f). At this point, we cannot discard WNT10B binding to or being expressed by pericytes; however, we consider that this possibility does not detract from the prospect that endfeet can receive WNT10B signals from BECs.

Despite the challenges of validating a secreted protein, we hope the reviewer will appreciate the remarkable robustness of our results, which consistently show WNT10B upregulation in capillaries of LPS mice across three different techniques (RNAseq, RNAscope, and immunohistochemistry). With RNAseq, *Wnt10b* is upregulated 2.8 times, with RNAscope – between 2.5 and 3 depending on the measure, and at the protein level, despite the known effects of post-translational modifications and the quantitative limitations of immunohistochemistry, we detected an upregulation of the WNT10B signal of 1.2-1.6 times. Every measure we took was statistically significant.

References

- Belmaati Cherkaoui M, Vacca O, Isabelle C, Boulay AC, Boulogne C, Gillet C, Barnier JV, Rendon A, Cohen-Salmon M, Vaillend C (2021) Dp71 contribution to the molecular scaffold anchoring aquaporine-4 channels in brain macroglial cells. *Glia* 69: 954-970
- Boulay AC, Saubamea B, Adam N, Chasseigneaux S, Mazare N, Gilbert A, Bahin M, Bastianelli L, Blugeon C, Perrin S *et al* (2017) Translation in astrocyte distal processes sets molecular heterogeneity at the gliovascular interface. *Cell Discov* 3: 17005
- Chai H, Diaz-Castro B, Shigetomi E, Monte E, Octeau JC, Yu X, Cohn W, Rajendran PS, Vondriska TM, Whitelegge JP *et al* (2017) Neural Circuit-Specialized Astrocytes: Transcriptomic, Proteomic, Morphological, and Functional Evidence. *Neuron* 95: 531-549 e539
- Diaz-Castro B, Gangwani MR, Yu X, Coppola G, Khakh BS (2019) Astrocyte molecular signatures in Huntington's disease. *Sci Transl Med* 11
- Ghazalpour A, Bennett B, Petyuk VA, Orozco L, Hagopian R, Mungrue IN, Farber CR, Sinsheimer J, Kang HM, Furlotte N *et al* (2011) Comparative analysis of proteome and transcriptome variation in mouse. *PLoS Genet* 7: e1001393
- Gokce O, Stanley GM, Treutlein B, Neff NF, Camp JG, Malenka RC, Rothwell PE, Fuccillo MV, Sudhof TC, Quake SR (2016) Cellular Taxonomy of the Mouse Striatum as Revealed by Single-Cell RNA-Seq. *Cell Rep* 16: 1126-1137
- Haenggi T, Soontornmalai A, Schaub MC, Fritschy JM (2004) The role of utrophin and Dp71 for assembly of different dystrophin-associated protein complexes (DPCs) in the choroid plexus and microvasculature of the brain. *Neuroscience* 129: 403-413
- Hayakawa K, Pham LD, Katusic ZS, Arai K, Lo EH (2012) Astrocytic high-mobility group box 1 promotes endothelial progenitor cell-mediated neurovascular remodeling during stroke recovery. *Proc Natl Acad Sci U S A* 109: 7505-7510
- Mishra A, Reynolds JP, Chen Y, Gourine AV, Rusakov DA, Attwell D (2016) Astrocytes mediate neurovascular signaling to capillary pericytes but not to arterioles. *Nat Neurosci* 19: 1619-1627
- Nicchia GP, Rossi A, Nudel U, Svelto M, Frigeri A (2008) Dystrophin-dependent and -independent AQP4 pools are expressed in the mouse brain. *Glia* 56: 869-876
- Ollivier M, Soto JS, Linker KE, Moyer SL, Jami-Alahmadi Y, Jones AE, Divakaruni AS, Kawaguchi R, Wohlschlegel JA, Khakh BS (2024) Crym-positive striatal astrocytes gate perseverative behaviour. *Nature* 627: 358-366
- Soto JS, Jami-Alahmadi Y, Chacon J, Moyer SL, Diaz-Castro B, Wohlschlegel JA, Khakh BS (2023) Astrocyte-neuron subproteomes and obsessive-compulsive disorder mechanisms. *Nature* 616: 764-773
- Tang D, Kang R, Zeh HJ, Lotze MT (2023) The multifunctional protein HMGB1: 50 years of discovery. *Nat Rev Immunol* 23: 824-841
- Tetorou K, Aghaeipour A, Singh S, Morgan JE, Muntoni F (2025) The role of dystrophin isoforms and interactors in the brain. *Brain* 148: 1081-1098
- Wang D, Eraslan B, Wieland T, Hallstrom B, Hopf T, Zolg DP, Zecha J, Asplund A, Li LH, Meng C *et al* (2019) A deep proteome and transcriptome abundance atlas of 29 healthy human tissues. *Mol Syst Biol* 15: e8503